# Topologically Regularized Data Embeddings

**Robin Vandaele, Bo Kang, Jefrey Lijffijt, Tijl De Bie, Yvan Saeys**
Ghent University, Gent, Belgium

## Abstract

Unsupervised feature learning often finds low-dimensional embeddings that capture the structure of complex data. For tasks for which prior expert topological knowledge is available, incorporating this into the learned representation may lead to higher quality embeddings. For example, this may help one to embed the data into a given number of clusters, or to accommodate for noise that prevents one from deriving the distribution of the data over the model directly, which can then be learned more effectively. However, a general tool for integrating different prior topological knowledge into embeddings is lacking. Although differentiable topology layers have been recently developed that can (re)shape embeddings into prespecified topological models, they have two important limitations for representation learning, which we address in this paper. First, the currently suggested topological losses fail to represent simple models such as clusters and flares in a natural manner. Second, these losses neglect all original structural (such as neighborhood) information in the data that is useful for learning. We overcome these limitations by introducing a new set of topological losses, and proposing their usage as a way for *topologically regularizing* data embeddings to naturally represent a prespecified model. We include thorough experiments on synthetic and real data that highlight the usefulness and versatility of this approach, with applications ranging from modeling high-dimensional single-cell data, to graph embedding.

## 1 Introduction

**Motivation** Modern data often arrives in complex forms that complicate their analysis. For example, high-dimensional data cannot be visualized directly, whereas relational data such as graphs lack the natural vectorized structure required by various machine learning models (Bhagat et al., 2011; Kazemi & Poole, 2018; Goyal & Ferrara, 2018). Representation learning aims to derive mathematically and computationally convenient representations to process and learn from such data. However, obtaining an effective representation is often challenging, for example, due to the accumulation of noise in high-dimensional biological expression data (Vandaele et al., 2021b). In other examples such as community detection in social networks, graph embeddings struggle to clearly separate communities with interconnections between them. In such cases, prior expert knowledge about the topological model may improve learning from, visualizing, and interpreting the data. Unfortunately, a general tool for incorporating prior topological knowledge in representation learning is lacking.

In this paper, we introduce such tool under the name of *topological regularization*. Here, we build on the recently developed differentiation frameworks for optimizing data to capture topological properties of interest (Brüel-Gabrielsson et al., 2020; Solomon et al., 2021; Carriere et al., 2021). Unfortunately, such *topological optimization* has been poorly studied within the context of representation learning. For example, the used *topological loss functions* are indifferent to any structure other than topological, such as neighborhood, which is useful for learning. Therefore, topological optimization often destructs natural and informative properties of the data in favor of the topological loss.

Our proposed method of *topological regularization effectively resolves this by learning an embedding representation that incorporates the topological prior*. As we will see in this paper, these priors can be directly postulated through topological loss functions. For example, if the prior is that the data lies on a circular model, we design a loss function that is lower whenever a more prominent cycle is present in the embedding. By extending the previously suggested topological losses to fit a wider set of models, we show that topological regularization effectively embeds data according to a variety of topological priors, ranging from clusters, cycles, and flares, to any combination of these.

**Related Work**   Certain methods that incorporate topological information into representation learning have already been developed. For example, Deep Embedded Clustering (Xie et al., 2016) simultaneously learns feature representations and cluster assignments using deep neural networks. Constrained embeddings of Euclidean data on spheres have also been studied by Bai et al. (2015). However, such methods often require an extensive development for one particular kind of input data and topological model. Contrary to this, incorporating topological optimization into representation learning provides a simple yet versatile approach towards combining data embedding methods with topological priors, that generalizes well to any input structure as long as the output is a point cloud.

Topological autoencoders (Moor et al., 2020) also combine topological optimization with a data embedding procedure. The main difference here is that the topological information used for optimization is obtained from the original high-dimensional data, and not passed as a prior. While this may sound as a major advantage—and certainly can be as shown by Moor et al. (2020)—obtaining such topological information heavily relies on distances between observations, which are often meaningless and unstable in high dimensions (Aggarwal et al., 2001). Furthermore, certain constructions such as the $\alpha$-*filtration* obtained from the Delaunay triangulation—which we will use extensively and is further discussed in Appendix A—are expensive to obtain from high-dimensional data (Cignoni et al., 1998), and are therefore best computed from the low-dimensional embedding.

Our work builds on a series of recent papers (Brüel-Gabrielsson et al., 2020; Solomon et al., 2021; Carriere et al., 2021), which showed that topological optimization is possible in various settings and developed the mathematical foundation thereof. However, studying the use of topological optimization for data embedding and visualization applications, as well as the new losses we develop therefore and the insights we derive from them in this paper, are to the best of our knowledge novel.

**Contributions**   We include a sufficient background on *persistent homology*—the method behind topological optimization—in Appendix A (note that all of its concepts important for this paper are summarized in Figure 1). We summarize the previous idea behind topological optimization of point clouds (Section 2.1). We introduce a new set of losses to model a wider variety of shapes in a natural manner (Section 2.2). We show how these can be used to topologically regularize embedding methods for which the output is a point cloud (Equation (1)). We include experiments on synthetic and real data that show the usefulness and versatility of topological regularization, and provide additional insights into the performance of data embedding methods (Section 3 & Appendix B). We discuss open problems in topological representation learning and conclude on our work (Section 4).

## 2   METHODS

The main purpose of this paper is to present a method to incorporate *prior topological knowledge* into a point cloud embedding $\boldsymbol{E}$ (dimensionality reduction, graph embedding, . . . ) of a data set $\mathbb{X}$. As will become clear below, these topological priors can be directly postulated through *topological loss functions* $\mathcal{L}_{\text{top}}$. Then, the goal is to find an embedding that minimizes a total loss function

$$\mathcal{L}_{\text{tot}}(\boldsymbol{E}, \mathbb{X}) \coloneqq \mathcal{L}_{\text{emb}}(\boldsymbol{E}, \mathbb{X}) + \lambda_{\text{top}}\mathcal{L}_{\text{top}}(\boldsymbol{E}), \tag{1}$$

where the loss function $\mathcal{L}_{\text{emb}}$ aims to preserve structural attributes of the original data, and $\lambda_{\text{top}} > 0$ controls the strength of *topological regularization*. Note that $\mathbb{X}$ itself is not required to be a point cloud, or reside in the same space as $\boldsymbol{E}$. This is especially useful for representation learning.

In this section, we focus on topological optimization of point clouds, that is, the loss function $\mathcal{L}_{\text{top}}$. The basic idea behind this recently introduced method—as presented by Brüel-Gabrielsson et al. (2020)—is illustrated in Section 2.1. We show that direct topological optimization may neglect important structural information such as neighborhoods, which can effectively be resolved through (1). Hence, as we will also see in Section 3, while representation learning may benefit from topological loss functions for incorporating prior topological knowledge, topological optimization itself may also benefit from other structural loss functions to represent the topological prior in a more truthful manner. Nevertheless, some topological models remain difficult to naturally represent through topological optimization. Therefore, we introduce a new set of topological loss functions, and provide an overview of how different models can be postulated through them in Section 2.2. Experiments and comparisons to topological regularization of embeddings through (1) will be presented in Section 3.

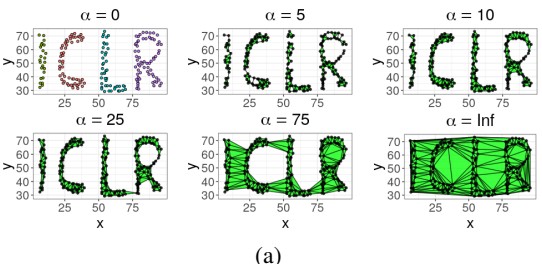 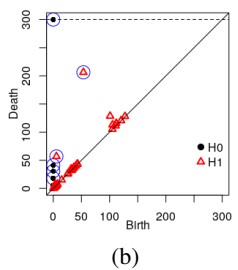

(a)                  (b)

Figure 1: The two basic concepts from persistent homology important for our method. (a) Persistent homology quantifies topological changes in a *filtration*, i.e., a changing sequence of *simplicial complexes* ordered by inclusion, parameterized by a time parameter $\alpha \in \mathbb{R}$. Intuitively, the parameter $\alpha$ determines the scale at which points become connected (see Definition A.0.1 in Appendix A for a formal definition of the $\alpha$-*complex*). Various topological holes are either born or die when $\alpha$ increases. The filtration starts with one connected component per point (*0-dimensional holes*), which can only merge together (resulting in their death) when including additional edges. For larger $\alpha$, we observe the birth of cycles (*1-dimensional holes*, here, 'white spaces enclosed by edges'), which get filled in (and thus die) when $\alpha$ increases further. Eventually, one connected component persists indefinitely. (b) Persistent homology is commonly visualized through *persistence diagrams*. Here, two are plotted on top of each other (H0 and H1). A tuple $(b, d)$ marks a hole—here a connected component (H0) or cycle (H1)—born at time $b$ that dies at (possibly infinite) time $d$ in the filtration. Prominently elevated points (encircled in blue) capture important topological properties, such as the four clusters 'I', 'C', 'L', and 'R' (H0), and the hole in the 'R', and between the 'C' and the 'L' (H1).

## 2.1 Background on Topological Optimization of Point Clouds

*Topological optimization* is performed through a *topological loss function* evaluated on one or more *persistence diagrams* (Barannikov, 1994; Carlsson, 2009). These diagrams—obtained through *persistent homology* as formally discussed in Appendix A—summarize all from the finest to coarsest topological holes (connected components, cycles, voids, . . . ) in the data, as illustrated in Figure 1.

While methods that learn from persistent homology are now both well developed and diverse (Pun et al., 2018), optimizing the data representation for the persistent homology thereof only gained recent attention (Brüel-Gabrielsson et al., 2020; Solomon et al., 2021; Carriere et al., 2021). Persistent homology has a rather abstract mathematical foundation within algebraic topology (Hatcher, 2002), and its computation is inherently combinatorial (Barannikov, 1994; Zomorodian & Carlsson, 2005). This complicates working with usual derivatives for optimization. To accommodate this, topological optimization makes use of Clarke subderivatives (Clarke, 1990), whose applicability to persistence builds on arguments from o-minimal geometry (van den Dries, 1998; Carriere et al., 2021). Fortunately, thanks to the recent work of Brüel-Gabrielsson et al. (2020) and Carriere et al. (2021), powerful tools for topological optimization have been developed for software libraries such as PyTorch and TensorFlow, allowing their usage without deeper knowledge about these subjects.

Topological optimization optimizes the data representation with respect to the topological information summarized by its persistence diagram(s) $\mathcal{D}$. We will use the approach by Brüel-Gabrielsson et al. (2020), where (birth, death) tuples $(b_1, d_1), (b_2, d_2), \ldots, (b_{|\mathcal{D}|}, d_{|\mathcal{D}|})$ in $\mathcal{D}$ are first ordered by decreasing persistence $d_k - b_k$. The points $(b, \infty)$, usually plotted on top of the diagram such as in Figure 1b, form the *essential part* of $\mathcal{D}$. The points with finite coordinates form the *regular part* of $\mathcal{D}$. In case of point clouds, one and only one topological hole, i.e., a connected component born at time $\alpha = 0$, will always persist indefinitely. Other gaps and holes will eventually be filled (Figure 1). Thus, we only optimize for the regular part in this paper. This is done through a *topological loss function*, which for a choice of $i \leq j$ (which, along with the dimension of topological hole, will specify our topological prior as we will see below), and a function $g : \mathbb{R}^2 \to \mathbb{R}$, is defined as

$$\mathcal{L}_{\mathrm{top}}(\mathcal{D}) := \sum_{k=i, d_k < \infty}^{j} g(b_k, d_k), \qquad \text{where } d_1 - b_1 \geq d_2 - b_2 \geq \ldots. \tag{2}$$

It turns out that for many useful definitions of $g$, $\mathcal{L}_{\mathrm{top}}(\mathcal{D})$ has a well-defined Clarke subdifferential with respect to the parameters defining the filtration from which the persistence diagram $\mathcal{D}$ is ob-

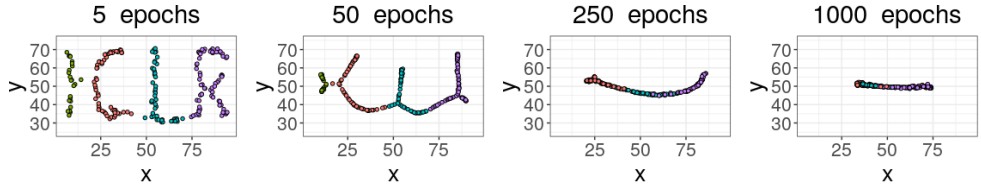

Figure 2: The data set in Figure 1a, optimized to have a low total 0-dimensional persistence. Points are colored according to their initial grouping along one of the four letters in the 'ICLR' acronym.

tained. In this paper, we will consistently use the $\alpha$-*filtration* as shown in Figure 1a (see Appendix A for its formal definition), and these parameters are entire point clouds (in this paper embeddings) $\boldsymbol{E} \in (\mathbb{R}^d)^n$ of size $n$ in the $d$-dimensional Euclidean space. $\mathcal{L}_{\text{top}}(\mathcal{D})$ can then be easily optimized with respect to these parameters through stochastic subgradient algorithms (Carriere et al., 2021).

As it directly measures the prominence of topological holes, we let $g : \mathbb{R}^2 \to \mathbb{R} : (b, d) \mapsto \mu(d - b)$ be proportional to the *persistence function*. By ordering the points by persistence, $\mathcal{L}_{\text{top}}$ is a *function of persistence*, i.e., it is invariant to permutations of the points in $\mathcal{D}$ (Carriere et al., 2021). The factor of proportionality $\mu \in \{1, -1\}$ indicates whether we want to *minimize* ($\mu = 1$) or *maximize* ($\mu = -1$) persistence, i.e, the prominence of topological holes. Thus, $\mu$ determines whether more clusters, cycles, ..., should be present ($\mu = -1$) or missing ($\mu = 1$). The loss (2) then reduces to

$$\mathcal{L}_{\text{top}}(\boldsymbol{E}) := \mathcal{L}_{\text{top}}(\mathcal{D}) = \mu \sum_{k=i, d_k < \infty}^{j} (d_k - b_k), \qquad \text{where } d_1 - b_1 \geq d_2 - b_2 \geq \dots. \quad (3)$$

Here, the data matrix $\boldsymbol{E}$ (in this paper the embedding) defines the diagram $\mathcal{D}$ through persistent homology of the $\alpha$-filtration of $\boldsymbol{E}$, and a persistence (topological hole) dimension to optimize for.

For example, consider (3) with $i = 2$, $j = \infty$, $\mu = 1$, restricted to 0-dimensional persistence (measuring the prominence of connected components) of the $\alpha$-filtration. Figure 2 shows the data from Figure 1 optimized for this loss function for various epochs. The optimized point cloud quickly resembles a single connected component for smaller numbers of epochs. This is the single goal of the current loss, which neglects all other structural structural properties of the data such as its underlying cycles (e.g., the circular hole in the 'R') or local neighborhoods. Larger numbers of epochs mainly affect the scale of the data. While this scale has an absolute effect on the total persistence, thus, the loss, the point cloud visually represents a single connected topological component equally well. We also observe that while local neighborhoods are preserved well during the first epochs simply by nature of topological optimization, they are increasingly distorted for larger numbers of epochs.

## 2.2 Newly Proposed Topological Loss Functions

In this paper, the prior topological knowledge incorporated into the point cloud data matrix embedding $\boldsymbol{E}$ is directly postulated through a topological loss function. For example, letting $\mathcal{D}$ be the 0-dimensional persistence diagram of $\boldsymbol{E}$ (H0 in Figure 1b), and $i = 2$, $j = \infty$, and $\mu = 1$ in (3), corresponds to the criterion that $\boldsymbol{E}$ should represents one closely connected component, as illustrated in Figure 2. Therefore, we often regard a *topological loss* as a *topological prior*, and vice versa.

Although persistent homology effectively measures the prominence of topological holes, topological optimization is often ineffective for representing such holes in a natural manner, for example, through sufficiently many data points. An extreme example of this are clusters, despite the fact that they are captured through the simplest form of persistence, i.e., 0-dimensional. This is shown in Figure 3. Optimizing the point cloud in Figure 3a for (at least) two clusters can be done by defining $\mathcal{L}_{\text{top}}(\boldsymbol{E})$ as in (3), letting $\mathcal{D}$ be the 0-dimensional persistence diagram of $\boldsymbol{E}$, $i = j = 2$, and $\mu = -1$. However, we observe that topological optimization simply displaces one point away from all other points (Figure 3b). Purely topologically, this is indeed a correct representation of two clusters.

To represent topological holes through more points, we propose topological optimization for the loss

$$\widetilde{\mathcal{L}}_{\text{top}}(\boldsymbol{E}) := \mathbb{E}\left[\mathcal{L}_{\text{top}}\left(\{\boldsymbol{x} \in \mathbf{S} : \mathbf{S} \text{ is a random sample of } \boldsymbol{E} \text{ with sampling fraction } f_{\mathcal{S}}\}\right)\right], \quad (4)$$

where $\mathcal{L}_{\text{top}}$ is defined as in (3). In practice, during each optimization iteration, $\widetilde{\mathcal{L}}_{\text{top}}$ is approximated by the mean of $\mathcal{L}_{\text{top}}$ evaluated over $n_{\mathcal{S}}$ random samples of $\boldsymbol{E}$. Figure 3 shows the result for a

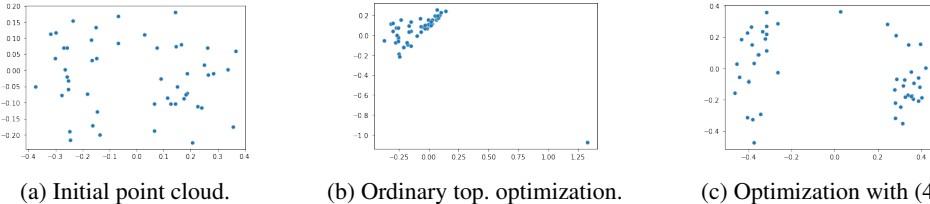

(a) Initial point cloud.  (b) Ordinary top. optimization.  (c) Optimization with (4).

Figure 3: A point cloud topologically optimized for (at least) two clusters, without and with sampling. Optimization with sampling encourages topological holes to be represented by more points.

sampling fraction $f_{\mathcal{S}} = 0.1$ and repeats $n_{\mathcal{S}} = 1$. Two larger clusters are now obtained. An added benefit of (4) is that optimization can be conducted significantly faster, as persistent homology is evaluated on smaller samples (a computational analysis can be found at the end of Appendix A).

In summary, various topological priors can now be formulated through topological losses as follows.

**$k$-dimensional holes**  Optimizing for $k$-dimensional holes ($k = 0$ for clusters), can generally be done through (3) or (4), by letting $\mathcal{D}$ be the corresponding $k$-dimensional persistence diagram. The terms $i$ and $j$ in the summation are used to control how many holes one wants to optimize for. Finally, $\mu$ can be chosen to either decrease ($\mu = 1$) or increase ($\mu = -1$) the prominence of holes.

**Flares**  Persistent homology is invariant to certain topological changes. For example, both a linear 'I'-structured model and a bifurcating 'Y'-structured model consist of one connected component, and no higher-dimensional holes. These models are indistinguishable based on the (persistent) homology thereof, even though they are topologically different in terms of their singular points.

Capturing such additional topological phenomena is possible through a refinement of persistent homology under the name of *functional persistence*, also well discussed and illustrated by Carlsson (2014). The idea is that instead of evaluating persistent homology on a data matrix $\boldsymbol{E}$, we evaluate it on a subset $\{\boldsymbol{x} \in \boldsymbol{E} : f(\boldsymbol{x}) \leq \tau\}$ for a well chosen function $f : \boldsymbol{E} \to \mathbb{R}$ and hyperparameter $\tau$.

Inspired by this approach, for a diagram $\mathcal{D}$ of a point cloud $\boldsymbol{E}$, we propose the topological loss

$$\widetilde{\mathcal{L}}_{\text{top}}(\boldsymbol{E}) := \mathcal{L}_{\text{top}}\left(\{\boldsymbol{x} \in \boldsymbol{E} : f_{\boldsymbol{E}}(\boldsymbol{x}) \leq \tau\}\right), \text{ informally denoted } \left[\mathcal{L}_{\text{top}}(\mathcal{D})\right]_{f_{\boldsymbol{E}}^{-1}]-\infty,\tau]}, \quad (5)$$

where $f$ is a real-valued function on $\boldsymbol{E}$, possibly dependent on $\boldsymbol{E}$—which changes during optimization—itself, $\tau$ a hyperparameter, and $\mathcal{L}_{\text{top}}$ is an ordinary topological loss as defined by (3). In particular, we will focus on the case where $f$ equals the scaled centrality measure on $\boldsymbol{E}$:

$$f_{\boldsymbol{E}} \equiv \mathcal{E}_{\boldsymbol{E}} := 1 - \frac{g_{\boldsymbol{E}}}{\max g_{\boldsymbol{E}}}, \text{ where } g_{\boldsymbol{E}}(\boldsymbol{x}) := \left\| \boldsymbol{x} - \frac{1}{|\boldsymbol{E}|} \sum_{\boldsymbol{y} \in \boldsymbol{E}} \boldsymbol{y} \right\|. \quad (6)$$

For $\tau \geq 1$, $\widetilde{\mathcal{L}}_{\text{top}}(\boldsymbol{E}) = \mathcal{L}_{\text{top}}(\boldsymbol{E})$. For sufficiently small $\tau > 0$, $\widetilde{\mathcal{L}}_{\text{top}}$ evaluates $\mathcal{L}_{\text{top}}$ on the points 'far away' from the mean in the center of $\boldsymbol{E}$. As we will see in the experiments, this is especially useful in conjunction with 0-dimensional persistence to optimize for *flares*, i.e., star-structured shapes.

**Combinations**  Naturally, through linear combination of loss functions, different topological priors can be combined, e.g., if we want the represented model to both be connected and include a cycle.

## 3  EXPERIMENTS

In this section, we show how our proposed topological regularization of data embeddings (1) leads to a powerful and versatile approach for representation learning. In particular, we show that

- embeddings benefit from prior topological knowledge through topological regularization;
- conversely, topological optimization may also benefit from incorporating structural information as captured through embedding losses, leading to more qualitative representations;
- subsequent learning tasks may benefit from prior expert topological knowledge.

In Section 3.1, we show how topological regularization improves standard PCA dimensionality reduction and allows better understanding of its performance when noise is accumulated over many dimensions. In Section 3.2, we present applications to high-dimensional biological single-cell trajectory data as well as graph embeddings. Quantitative results are discussed in Section 3.3.

Topological optimization was performed in Pytorch on a machine equipped with an Intel® CoreTM i7 processor at 2.6GHz and 8GB of RAM, using code adapted from Brüel-Gabrielsson et al. (2020). Tables 1 & 2 summarize data sizes, hyperparameters, losses, and optimization times. Appendix B presents supplementary experiments that show topological regularization is consistent with earlier analysis of the Harry Potter network (Section B.1), a domain application in cell trajectory inference (Section B.2), and how topological regularization reacts to different topological losses and hyperparameters (Section B.3). Code is available on `github.com/robinvndaele/topembedding`.

## 3.1 SYNTHETIC DATA

We sampled 50 points uniformly from the unit circle in $\mathbb{R}^2$. We then added 500-dimensional noise to the resulting data matrix $\boldsymbol{X}$, where the noise in each dimension is sampled uniformly from $[-0.45, 0.45]$. Since the additional noisy features are irrelevant to the topological (circular) model, an ideal projection embedding $\boldsymbol{X}$ is its restriction to its first two data coordinates (Figure 4a).

However, it is probabilistically unlikely that that the irrelevant features will have a zero contribution to a PCA embedding of the data (Figure 4b). Measuring the feature importance of each feature as the sum of its two absolute contributions (the loadings) to the projection, we observe that most of the 498 irrelevant features have a small nonzero effect on the PCA embedding (Figure 5). Intuitively, each added feature slightly shifts the projection plane away from the plane spanned by the first two coordinates. As a result, the circular hole is less prominent in the PCA embedding of the data. Note that *we observed this to be a notable problem for 'small n large p' data sets*, as similar to other machine learning models, and as also recently studied by Vandaele et al. (2021a), more data can significantly accommodate for the effect of noise and result in a better embedding model on its own.

We can regularize this embedding using a topological loss function $\mathcal{L}_{\mathrm{top}}$ measuring the persistence of the most prominent 1-dimensional hole in the embedding ($i = j = 1$ in (3)). For a simple Pytorch compatible implementation, we used $\mathcal{L}_{\mathrm{emb}}(\boldsymbol{W}, \boldsymbol{X}) \coloneqq \mathrm{MSE}\left(\boldsymbol{XWW}^T, \boldsymbol{X}\right)$, as to minimize the reconstruction error between $\boldsymbol{X}$ and its linear projection obtained through $\boldsymbol{W}$. To this, we added the loss $10^4\mathcal{L}_\perp(\boldsymbol{W})$, where $\mathcal{L}_\perp(\boldsymbol{W}) \coloneqq \|\boldsymbol{W}^T\boldsymbol{W} - \boldsymbol{I}_2\|_2$ encourages orthonormality of the matrix $\boldsymbol{W}$ for which we optimize, initialized with the PCA-loadings. The resulting embedding is shown in Figure 4d (here $\mathcal{L}_\perp(\boldsymbol{W}) \sim 0.03$), which better captures the circular hole. Furthermore, we see that irrelevant features now more often contribute less to the embedding according to $\boldsymbol{W}$ (Figure 5).

For comparison, Figure 4c shows the optimized embedding without the reconstruction loss $\mathcal{L}_{\mathrm{emb}}$. The loss $\mathcal{L}_\perp$ is still included (here $\mathcal{L}_\perp(\boldsymbol{W}) \sim 0.2$). From this and also from Figure 5, we observe that $\boldsymbol{W}$ struggles more to converge to the correct projection, resulting in a less prominent hole.

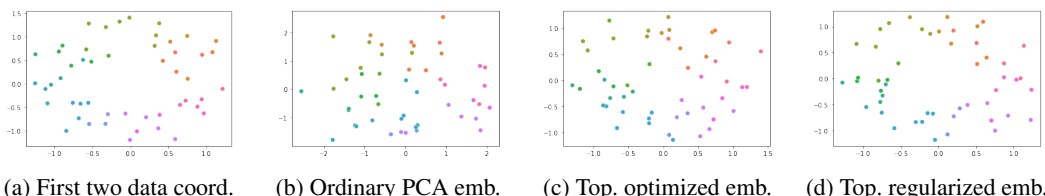

| (a) First two data coord. | (b) Ordinary PCA emb. | (c) Top. optimized emb. | (d) Top. regularized emb. |

Figure 4: Various representations of the synthetic data, colored by ground truth circular coordinates.

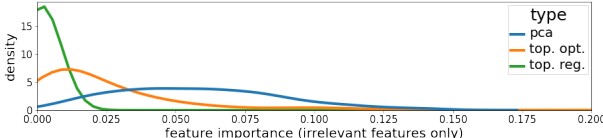

Figure 5: Feature importance densities of the 498 irrelevant features in the PCA embedding (blue), the top. optimized PCA embedding (orange), and the top. regularized PCA embedding (green).

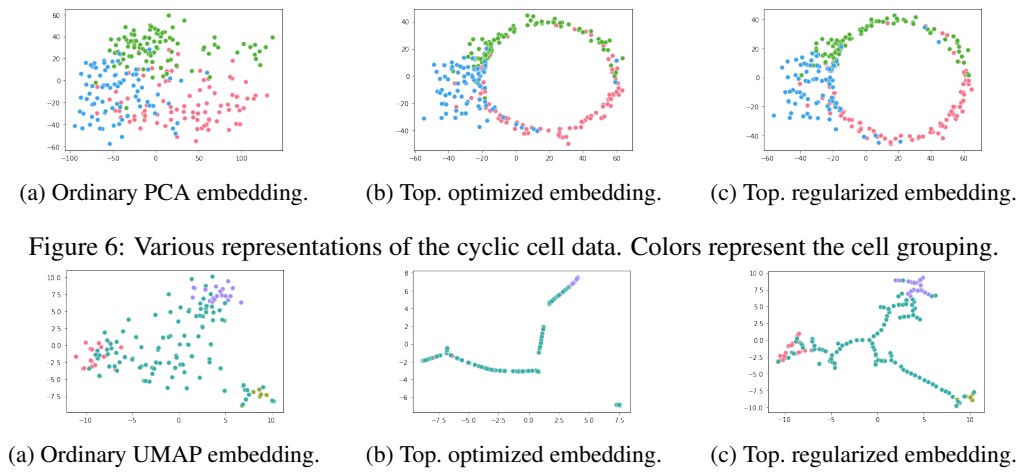

(a) Ordinary PCA embedding.    (b) Top. optimized embedding.    (c) Top. regularized embedding.

Figure 6: Various representations of the cyclic cell data. Colors represent the cell grouping.

(a) Ordinary UMAP embedding.    (b) Top. optimized embedding.    (c) Top. regularized embedding.

Figure 7: Various representations of the bifurcating cell data. Colors represent the cell grouping.

## 3.2 REAL DATA

**Circular Cell Trajectory Data** We considered a single-cell trajectory data set of 264 cells in a 6812-dimensional gene expression space (Cannoodt et al., 2018; Saelens et al., 2019). This data can be considered a snapshot of the cells at a fixed time. The ground truth is a circular model connecting three cell groups through cell differentiation (see also Section B.2). It has been shown by Vandaele (2020) that real single-cell data derived from such model are difficult to embed in a circular manner.

To explore this, we repeated the experiment with the same losses as in Section 3.1 on this data, where the (expected) topological loss is now modified through (4) with $f_{\mathcal{S}} = 0.25$, and $n_{\mathcal{S}} = 1$. From Figure 6a, we see that while the ordinary PCA embedding does somehow respect the positioning of the cell groups (marked by their color), it indeed struggles to embed the data in a manner that visualizes the present cycle. However, as shown in Figure 6c, by topologically regularizing the embedding (here $\mathcal{L}_{\perp}(\boldsymbol{W}) \sim 4e^{-3}$) we are able to embed the data much better in a circular manner.

Figure 6b shows the optimized embedding without the loss $\mathcal{L}_{\mathrm{emb}}$. The embedding is similar to the one in Figure 6c (here $\mathcal{L}_{\perp}(\boldsymbol{W}) \sim 4e^{-3}$), with the pink colored cell group slightly more dispersed.

**Bifurcating Cell Trajectory Data** We considered a second cell trajectory data set of 154 cells in a 1770-dimensional expression space (Cannoodt et al., 2018). The ground truth here is a bifurcating model connecting four different cell groups through cell differentiation. However, this time we used the UMAP loss for the embeddings. We used a topological loss $\mathcal{L}_{\mathrm{top}} \equiv \mathcal{L}_{\mathrm{conn}} - \mathcal{L}_{\mathrm{flare}}$, where $\mathcal{L}_{\mathrm{conn}}$ measures the total (sum of) finite 0-dimensional persistence in the embedding to encourage connectedness of the representation, and $\mathcal{L}_{\mathrm{flare}}$ is as in (5), measuring the persistence of the third most prominent 0-dimensional hole in $\{\boldsymbol{y} \in \boldsymbol{E} : \mathcal{E}_{\boldsymbol{E}}(\boldsymbol{y}) \leq 0.75\}$, where $\mathcal{E}_{\boldsymbol{E}}$ is as in (6). Thus, $\mathcal{L}_{\mathrm{flare}}$ is used to optimize for a 'flare' with (at least) three clusters away from the embedding mean. We observe that while the ordinary UMAP embedding is more dispersed (Figure 7a), the topologically regularized embedding is more constrained towards a connected bifurcating shape (Figure 7c).

For comparison, we conducted topological optimization for the loss $\mathcal{L}_{\mathrm{top}}$ of the initialized UMAP embedding without the UMAP embedding loss. The resulting embedding is now more fragmented (Figure 7b). We thus see that topological optimization may also benefit from the embedding loss.

**Graph Embedding** The topological loss in (1) can be evaluated on any embedding, and does not require a point cloud as original input. We can thus use topological regularization for embedding a graph $G$, to learn a representation of the nodes of $G$ in $\mathbb{R}^d$ that well respects properties of $G$.

To explore this, we considered the Karate network (Zachary, 1977), a well known and studied network within graph mining that consists of two different communities. The communities are represented by two key figures (John A. and Mr. Hi), as shown in Figure 8a. To embed the graph, we used a DeepWalk variant adapted from Dagar et al. (2020). While the ordinary DeepWalk em-

bedding (Figure 8b) well respects the ordering of points according to their communities, the two communities remained close to each other. We thus regularized this embedding for (at least) two clusters using the topological loss as defined by (4), where $\mathcal{L}_{\text{top}}$ measures the persistence of the second most prominent 0-dimensional hole, and $f_{\mathcal{S}} = 0.25$, $n_{\mathcal{S}} = 10$. The resulting embedding (Figure 8d) now nearly perfectly separates the two ground truth communities present in the graph.

Topological optimization of the ordinary DeepWalk embedding with the same topological loss function but without the DeepWalk loss function creates some natural community structure, but results in a few outliers (Figure 8c). Thus, although our introduced loss (4) enables more natural topological modeling to some extent, we again observe that using this in conjunction with embedding losses, i.e., our proposed method of topological regularization, leads to the best qualitative results.

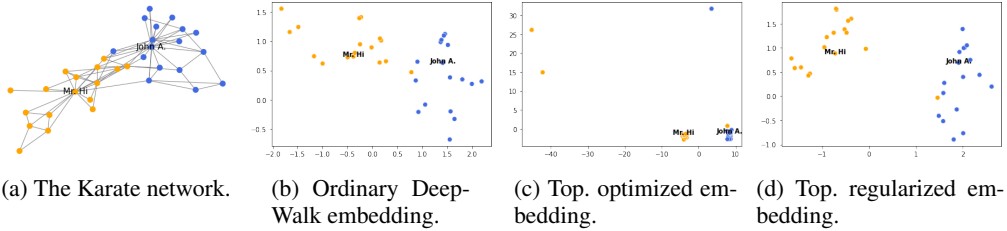

| (a) The Karate network. | (b) Ordinary Deep-Walk embedding. | (c) Top. optimized embedding. | (d) Top. regularized embedding. |

Figure 8: The Karate network and various of its embeddings.

### 3.3 QUANTITATIVE EVALUATION

Table 3 summarizes the embedding and topological losses we obtained for the ordinary embeddings, the topologically optimized embeddings (initialized with the ordinary embeddings, but not using the embedding loss), as well as for the topologically regularized embeddings. As one would expect, topological regularization balances the embedding losses between the embedding losses of the ordinary and topologically optimized embeddings. More interestingly, topological regularization may actually result in a more optimal, i.e., lower topological loss than topological optimization only, here in particular for the synthetic cycle data and Harry Potter graph. This suggest that combining topological information with other structural information may facilitate convergence to the correct embedding model, as we also qualitatively confirmed for these data sets (see also Section B.1). We also observe that there are more significant differences in the obtained topological losses than in the embedding losses with and without regularization. This suggests that the optimum region for the embedding loss may be somewhat flat with respect to the corresponding region for the topological loss. Thus, slight shifts in the local embedding optimum, e.g., as caused by noise, may result in much worse topological embedding models, which can be resolved through topological regularization.

Table 1: Summarization of the data, hyperparameters and optimization times for our experiments. The size format is $\#points \times \#dimensions$ for point clouds, and $\#vertices \times \#edges$ for graphs.

| data | size | method | lr | epochs | $\lambda_{\text{top}}$ | $t$ w/o top | $t$ with top |
|---|---|---|---|---|---|---|---|
| Synthetic Cycle | $50 \times 500$ | PCA | 1e-1 | 500 | 1e1 | <1s | 5s |
| Cell Cycle | $264 \times 6812$ | PCA | 5e-4 | 1000 | 1e2 | <1s | 35s |
| Cell Bifurcating | $154 \times 1770$ | UMAP | 1e-1 | 100 | 1e1 | <1s | 8s |
| Karate | $34 \times 78$ | DeepWalk | 1e-2 | 50 | 5e1 | 29s | 29s |
| Harry Potter | $58 \times 217$ | InnerProd | 1e-1 | 100 | 1e-1 | 36s | 34s |

Table 2: Summary of the topological losses computed from persistence diagrams $\mathcal{D}$ with points $(b_k, d_k)$ ordered by persistence $d_k - b_k$. Note that for 0-th dimensional homology diagrams $d_1 = \infty$.

| data | top. loss function | dimension of hole | $f_{\mathcal{S}}$ | $n_{\mathcal{S}}$ |
|---|---|---|---|---|
| Synthetic Cycle | $-(d_1 - b_1)$ | 1 | N/A | N/A |
| Cell Cycle | $-(d_1 - b_1)$ | 1 | 0.25 | 1 |
| Cell Bifurcating | $\sum_{k=2}^{\infty}(d_k - b_k) - [d_3 - b_3]_{\mathcal{E}_E^{-1}]-\infty, 0.75]}$ | 0 - 0 | N/A | N/A |
| Karate | $-(d_2 - b_2)$ | 0 | 0.25 | 10 |
| Harry Potter | $-(d_1 - b_1)$ | 1 | N/A | N/A |

Table 3: Embedding/reconstruction and topological losses of the final embeddings. Lowest in bold.

| data | embedding loss | | | topological loss | | |
|---|---|---|---|---|---|---|
| | ord. emb. | top. opt. | top. reg. | ord. emb. | top. opt. | top. reg. |
| Synthetic Cycle | $\mathbf{6.3e^{-2}}$ | $6.7e^{-2}$ | $6.6e^{-2}$ | $-0.15$ | $-0.35$ | $\mathbf{-0.75}$ |
| Cell Cycle | $\mathbf{6.70}$ | $7.00$ | $6.99$ | $-13.4$ | $\mathbf{-50.9}$ | $-49.7$ |
| Cell Bifurcating | $\mathbf{8576}$ | $9933$ | $8871$ | $117$ | $\mathbf{23}$ | $63$ |
| Karate | $\mathbf{2006}$ | N/A | $2112$ | $-1.3$ | $\mathbf{-28.5}$ | $-2.4$ |
| Harry Potter | $\mathbf{0.20}$ | $1.11$ | $0.23$ | $-0.82$ | $-2.37$ | $\mathbf{-3.05}$ |

Table 4: Embedding performance evaluations for label prediction. Highest in bold.

| data | metric | ord. emb. | top. opt. | top. reg. |
|---|---|---|---|---|
| Synthetic Cycle | $r^2$ | $0.56 \pm 0.47$ | $0.77 \pm 0.24$ | $\mathbf{0.85 \pm 0.14}$ |
| Cell Cycle | accuracy | $0.79 \pm 0.07$ | $0.79 \pm 0.07$ | $\mathbf{0.81 \pm 0.07}$ |
| Cell Bifurcating | accuracy | $0.79 \pm 0.08$ | $0.81 \pm 0.07$ | $\mathbf{0.82 \pm 0.08}$ |
| Karate | accuracy | $\mathbf{0.97 \pm 0.08}$ | $0.91 \pm 0.14$ | $\mathbf{0.97 \pm 0.08}$ |

We evaluated the quality of the embedding visualizations presented in this section, by assessing how informative they are for predicting ground data truth labels. For the Synthetic Cycle data, these labels are the 2D coordinates of the noise-free data on the unit circle in $\mathbb{R}^2$, and we used a multi-ouput support vector regressor model. For the cell trajectory data and Karate network, we used the ground truth cell groupings and community assignments, respectively, and a support vector machine model. All points in the 2D embeddings were then split into 90% points for training and 10% for testing. Consecutively, we used 5-fold CV on the training data to tune the regularization hyperparameter $C \in \{1e-2, 1e-1, 1, 1e1, 1e2\}$. Other settings were the default from SCIKIT-LEARN. The performance of the final tuned and trained model was then evaluated on the test data, through the $r^2$ coefficient of determination for the regression problem, and the accuracy for all classification problems. The averaged test performance metrics and their standard deviations, obtained over 100 random train-test splits, are summarized in Table 4. Although topological regularization consistently leads to more informative visualization embeddings, quantitative differences can be minor, as from the figures in this section we observe that most local similarities are preserved with regularization.

## 4 DISCUSSION AND CONCLUSION

We proposed a new approach for representation learning under the name of *topological regularization*, which builds on the recently developed differentiation frameworks for topological optimization. This led to a versatile and effective way for embedding data according to prior expert topological knowledge, directly postulated through (some newly introduced) topological loss functions.

A clear limitation of topological regularization is that prior topological knowledge is not always available. How to select the best from a list of priors is thus open to further research (see also Section B.3). Furthermore, designing topological loss functions currently requires some understanding of persistent homology. It would be useful to study how to facilitate that design process for lay users. From a foundational perspective, our work provides new research opportunities into extending the developed theory for topological optimization (Carriere et al., 2021) to our newly introduced losses and their integration into data embeddings. Finally, topological optimization based on combinatorial structures other than the $\alpha$-complex may be of theoretical and practical interest. For example, point cloud optimization based on graph-approximations such as the minimum spanning tree (Vandaele et al., 2021b), or varying the functional threshold $\tau$ in the loss (5) alongside the filtration time (Chazal et al., 2009), may lead to natural topological loss functions with fewer hyperparameters.

Nevertheless, through our approach, we already provided new and important insights into the performance of embedding methods, such as their potential inability to converge to the correct topological model due to the flatness of the embedding loss near its (local) optimum, with respect to the topological loss. Furthermore, we showed that including prior topological knowledge provides a promising way to improve consecutive—even non-topological—learning tasks (see also Section B.2). In conclusion, topological regularization enables both improving and better understanding representation learning methods, for which we provided and thoroughly illustrated the first directions in this paper.

ACKNOWLEDGMENTS

The research leading to these results has received funding from the European Research Council under the European Union's Seventh Framework Programme (FP7/2007-2013) (ERC Grant Agreement no. 615517), and under the European Union's Horizon 2020 research and innovation programme (ERC Grant Agreement no. 963924), from the Flemish Government under the "Onderzoeksprogramma Artificiële Intelligentie (AI) Vlaanderen" programme, and from the FWO (project no. G091017N, G0F9816N, 3G042220).

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

## A    APPENDIX: INTRODUCTION TO PERSISTENT HOMOLOGY

In this part of the appendix, we provide a visual introduction to *persistent homology* and *persistence diagrams* that targets a broad machine learning audience. Although the fundamental results leading to persistent homology are mathematically abstract (Hatcher, 2002), its purpose and what it computes are rather easy to visualize.

A custom case in data science for illustrating how persistent homology works, is that we have a point cloud data set $X$ embedded in a Euclidean space $\mathbb{R}^d$ (in the main paper, $X$ is our embedding $E$). In particular, we will use the point cloud $X$ that is visualized in the top left plot in Figure 9 as a working example. Note that this is also the data set resembling the 'ICLR' acronym from the main paper, and that Figure 9 is identical to Figure 1a from the main paper. The task is now to infer topological properties of the model underlying $X$, by means of persistent homology.

**Simplicial Complexes**    Looking at $X$ (Figure 9, Top Left), the only topological information that can be deduced from it, is that it is a set of points. This is because no two point clouds of the same size can be topologically distinguished (at least not according to any metric defined on them, which we assume to be the Euclidean distance metric here). Indeed, the displacement of isolated points in the Euclidean space, and more generally continuous stretchings and bendings, correspond to *homeomorphisms*, which are functions between topological spaces that preserve all topological information.

A partial solution to this can be obtained by constructing a *simplicial complex* from $X$. In general, a simplicial complex can be seen as a generalization of a graph, where apart from nodes (0-simplices) and edges (1-simplices), it may also include triangles (2-simplices), tetrahedra (3-simplices), and so on. More specifically, the two defining properties of a simplicial complex $\Delta$ are that

- $\Delta$ is a set of finite subsets of some given set $\mathcal{S}$;
- If $\sigma' \subseteq \sigma \in \Delta$, then $\sigma' \in \Delta$.

Each element $\sigma$ in such a simplicial complex is called a *face* or *simplex*. If $|\sigma| = k + 1$, it is also called a $k$-simplex, and $k$ is called the *dimension* of $\sigma$.

One should note that this is formally the definition of an *abstract* simplicial complex, and the term simplicial complex commonly refers to a *geometrical realization* of such a complex. Such realization can be seen as a continuous drawing of the complex in some Euclidean space $\mathbb{R}^d$, such that the intersection of two simplices $\sigma$ and $\sigma'$ corresponds to a common face of $\sigma$ and $\sigma'$. This is similar to how a planar drawing of a graph (without intersecting edges) or a metric graph is a geometric realization of a graph (Vandaele et al., 2021c). However, for simplicial complexes we also 'fill in' the 2-simplices, 3-simplices, ..., which is usually how we visualize (abstract) simplicial complexes,

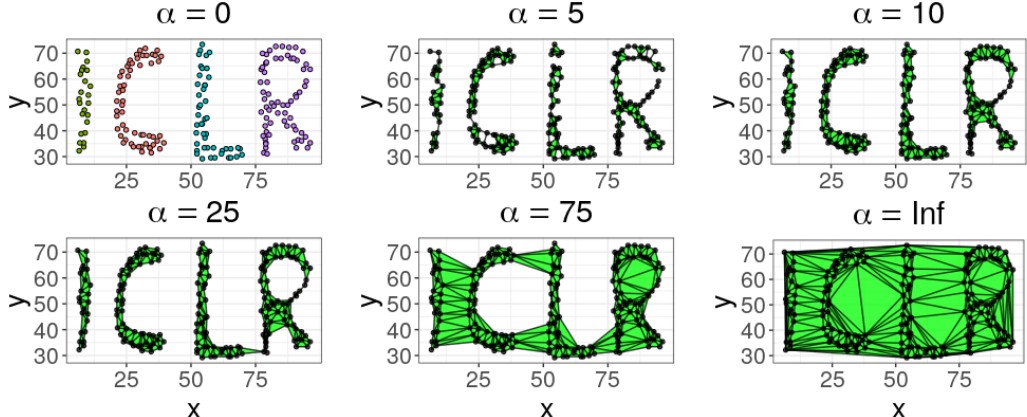

Figure 9: An example of six simplicial complexes $\Delta_\alpha(X)$ in the $\alpha$-filtration filtration of a point cloud data set $X$ resembling the 'ICLR' conference acronym in the Euclidean plane. $\alpha = \infty$ is informally used to denote the complex that contains all simplices, which here equals the Delaunay triangulation, and will necessarily be identical to $\Delta_\alpha(X)$ for some finite $\alpha \in \mathbb{R}_{>0}$.

such as in Figure 9. Sometimes the term 'simplex' is only used to refer to geometric realizations of their discrete counterparts, and the discrete counterparts are only referred to as 'faces'. For the purpose of this paper, it is not an issue for one to mix up this terminology, i.e., to identify simplicial complexes with their discrete counterparts, and we will often proceed to do this.

The most important thing to be aware of, is that (persistent) homology is concerned with topological properties of the 'continuous versions' (geometric realizations) of simplicial complexes, which are computed through their 'discrete' (abstract) counterparts. Compare this to how the connectedness of a graph (as a graph, or thus, a finite combinatorial structure,) determines the connectedness of any of its geometric realizations (as a topological space containing uncountably many points).

**The $\alpha$-complex** When given a point cloud $X$ (or in this paper an embedding $E$), we consider the $\alpha$-*complex* $\Delta_\alpha(X)$ at 'time' $\alpha \in \mathbb{R}_{\geq 0}$. This complex summarize topological information from the continuous model underlying the discrete point cloud $X$ at the 'scale' determined by $\alpha$. It is a *subcomplex* of the *Delaunay triangulation* (Delaunay et al., 1934) of $X$, as shown in Figure 9. The Delaunay triangulation equals the simplicial complex at the informally used time $\alpha = \infty$. Note that the definition of the $\alpha$-complex as a subset of the Delaunay triangulation below (Boissonnat et al., 2018) can be more difficult to grasp than the definition of other simplicial complexes, such as the *Vietoris-Rips complex* (Otter et al., 2017), that are not used in this paper. However, the exact definition is not required to understand the basic ideas presented in the main paper, and is mainly included for completeness. What is most important is that the $\alpha$-complex aims to quantify topological properties underlying $X$, as is illustrated in Figure 9 and will be further discussed below. For a good overview and visualization of how the $\alpha$-filtration is constructed, we refer the interested reader to The GUDHI Project (2021), from which we obtained the following definition.

**Definition A.0.1** ($\alpha$-complex). *Let $X$ be a point cloud in the Euclidean space $\mathbb{R}^d$, and $\Delta$ the Delaunay triangulation of $X$. For $\alpha = 0$, the $\alpha$-complex $\Delta_{\alpha=0}(X)$ of $X$ simply equals the set of points of $X$ (thus a simplicial complex with only 0-simplices). More generally, to every simplex $\sigma$ in $\Delta$, we assign a time value $\alpha$, which equals the square of the circumradius of $\sigma$ if its circumsphere contains no other vertices than those in $\sigma$, in which case $\sigma$ is said to be* Gabriel, *and as the minimum time values of the $(|\sigma| + 1)$-simplices containing $\sigma$ that make it not Gabriel otherwise. For $\alpha > 0$, the $\alpha$-complex $\Delta_\alpha(X)$ of $X$ is now defined as the set of simplices in $\Delta$ with time value at most $\alpha$.*

**Topological Holes and Betti Numbers** Once we have constructed an $\alpha$-complex $\Delta_\alpha(X)$ from $X$, we can now infer topological properties that are more interesting than $X$ being a set of isolated points. For example, in Figure 9, we see that $\Delta_{10}(X)$ captures many important topological properties of the model underlying $X$, which consists of four disconnected components (one for each of the letters 'I', 'C', 'L', and 'R' in the conference acronym), and also captures the circular hole that is present in the letter 'R'. *Homology* exactly quantifies such information by associating *Betti numbers* $\beta_k$ to the simplicial complex. $\beta_k$ corresponds to how many $k$-*dimensional holes* there are in the complex. In this sense, a 0-dimensional hole correspond to a gap between two components, and $\beta_0$ equals the number of connected components. A 1-dimensional hole corresponds to a loop (which can be seen as a circle, ring, or a handle of a coffee mug), and a 2-dimensional hole corresponds to a void (which can be seen as the inside of a balloon). In general, an $n$-dimensional hole corresponds to the interior of a $n$-sphere in $\mathbb{R}^{n+1}$. Note that no $n$-dimensional holes can occur in the Euclidean space $\mathbb{R}^d$ whenever $d \leq n$, and therefore (as well as for computational purposes) one often restricts the dimension $k$ of the simplicial complexes that are constructed from the data.

**Filtrations and $\alpha$-Filtrations** The difficulty lies in pinpointing an exact value for $\alpha$ for which $\Delta_\alpha(X)$ truthfully captures the topological properties of the model underlying $X$. For example, $\Delta_{25}(X)$ still captures the hole in the 'R', but connects the points representing the letters 'L' and 'R' into one connected component. $\Delta_{75}(X)$ captures only one connected component, but also captures a prominent hole that is composed between the letters 'C' and 'R'. This larger hole can also be seen as a particular topological characteristic of the underlying model, though on a different scale. This is where 'persistent' homology comes into play. Rather than inferring these topological properties (holes) for one particular simplicial complex, the task of persistent homology is to track the change of these topological properties across a varying sequence of simplicial complexes

$$\Delta_0 \subseteq \Delta_1 \subseteq \ldots \subseteq \Delta_n,$$

which is termed a *filtration*. The definition of an $\alpha$-simplex directly gives rise to such a filtration, termed the $\alpha$-*filtration*:

$$(\Delta_\alpha(\boldsymbol{X}))_{\alpha \in \mathbb{R}_{\geq 0}}.$$

Indeed, for $\alpha < \alpha'$, it follows directly from the definition of $\alpha$-complexes that $\Delta_\alpha(\boldsymbol{X}) \subseteq \Delta_{\alpha'}(\boldsymbol{X})$. Note that in practice, for finite data, the simplicial complex across this filtration changes for only finitely many time values $\alpha$, and we may indeed regard this filtration to be of the (finite) form $\Delta_0 \subseteq \Delta_1 \subseteq \ldots \subseteq \Delta_n$. The time parameter $\alpha$ is considered to parameterize the filtration, and is thus also termed the *filtration time*.

**Persistent Homology**   When given a filtration parameterized by a time parameter $t$ (e.g., $t = \alpha$), one may observe changes in the topological holes when $t$ increases. For example, as illustrated in Figure 9, different connected components may become connected through edges, or cycles may either appear or disappear. When a topological hole comes to existence in the filtration, we say that it is *born*. Vice versa, we say that a topological hole *dies* when it disappears. The filtration time at which these events occur are called the *birth time* and *death time*, respectively. Simply put, *persistent homology* tracks the birth and death times of topological holes across a filtration. 'Homology' refers to the topic within algebraic topology that considers the study and computation of these holes in topological models, and rests on (often abstract) concepts from mathematical fields such as group theory and linear algebra. As it may distract one from the main purpose of persistent homology we aim to introduce in this paper, we will not provide an introduction to these topics here, but we refer the interested reader to Hatcher (2002). 'Persistent' refers to the fact that one tracks homology, or thus topological holes, across an increasing filtration time parameter, and that *persistent holes—those that remain present for many consecutive time values $t$—are commonly regarded to be informative for the (underlying) topological model in the data*. This is the interpretation of persistent homology maintained for topological regularization, as well as for designing topological loss functions, in the main paper. However, more recent work shows that also finer topological holes that persist for short time intervals can be effective for machine learning applications, for example, for protein structure classification (Pun et al., 2018).

**Persistence Diagrams**   *Persistence diagrams* are a tool to capture and visualize the information quantified through persistent homology of a filtration. Formally, a persistence diagram is a set

$$\mathcal{D}_k = \underbrace{\{(t_{a_i}, \infty) : 1 \leq i \leq N\}}_{=: \mathcal{D}_k^{\mathrm{ess}}} \cup \underbrace{\{(t_{b_j}, t_{d_j}) : 1 \leq j \leq M \wedge t_{b_j} < t_{d_j}\}}_{=: \mathcal{D}_k^{\mathrm{reg}}} \cup \{(x, x) : x \in \mathbb{R}\},$$

where $t_{a_i}, t_{b_j}, t_{d_j}$, $1 \leq i \leq N$, $1 \leq j \leq M$, correspond to the birth and death times of $k$-dimensional holes across the filtration. Points $(t_{a_i}, \infty)$, are usually displayed on top of the diagram. They correspond to holes that never die across the filtration, and form the *essential part* of the persistence diagram. In the case of $\alpha$-filtrations, one always has that $\mathcal{D}_0^{\mathrm{ess}} = \{(0, \infty)\}$, and $\mathcal{D}_k^{\mathrm{ess}} = \emptyset$ for $k \geq 1$. This is because eventually, the simplicial complex in the filtration will consist of one connected component that never dies, and any higher dimensional hole will be 'filled in', as illustrated by Figure 9. Hence, during topological regularization of data embeddings, and as discussed in the main paper, we cannot topologically optimize for the essential part of any persistence diagram of our embedding. The points $(t_{b_j}, t_{d_j})$ in $\mathcal{D}_k$ with finite coordinates $t_{b_j} < t_{d_j}$ form the *regular part* $D_k^{\mathrm{reg}}$ of the persistence diagram. These are the points for which we optimize with topological regularization. Finally, the diagonal is included in a persistence diagram, as to allow for a well-defined distance metric between persistence diagram, termed the *bottleneck distance*.

**Definition A.0.2** *(bottleneck distance). Let $\mathcal{D}$ and $\mathcal{D}'$ be two persistence diagrams. The* bottleneck distance *between them is defined as*

$$d_{\mathrm{b}}(\mathcal{D}, \mathcal{D}') := \inf_{\varphi} \sup_{x} \|x - \varphi(x)\|_\infty \in \mathbb{R} \cup \{\infty\},$$

*where $\varphi$ ranges over all bijections from $\mathcal{D}$ to $\mathcal{D}'$, and $x$ ranges over all points in $\mathcal{D}$. By convention, we let $\infty - \infty = 0$ when calculating the distance between two diagram points. Since persistence diagrams include the diagonal by definition, $|\mathcal{D}| = |\mathcal{D}'| = |\mathbb{R}|$. Thus, $d_{\mathrm{b}}(\mathcal{D}, \mathcal{D}')$ is well-defined, as unmatched points in the essential or regular part of a diagram can always be matched to the diagonal.*

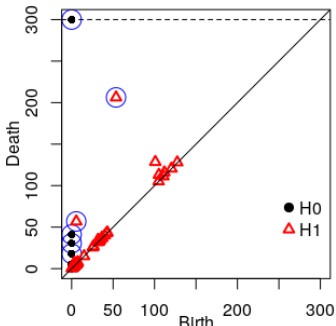

Figure 10: The diagrams $\mathcal{D}_0$ (black points, H0) and $\mathcal{D}_1$ (red points, H1) for the $\alpha$-filtration filtration of the point cloud data set in Figure 9, plotted on top of each other. The six highly elevated points (encircled in blue) identify the presence of four connected components (H0), corresponding to the 'I', 'C', 'L', and 'R' letters in the conference acronym, and two cycles (H1), corresponding to the hole in the letter 'R' (born earlier) and composed between the letters 'R' and 'C' (born later).

The bottleneck distance commonly justifies the use of persistent homology as a stable method for quantifying topological information in data, i.e., robust to small data permutations (Oudot, 2015). Note that although the bottleneck distance is not used in the main paper, we can also conduct topological optimization with respect to this metric (Carriere et al., 2021).

Figure 10 shows the persistence diagrams $\mathcal{D}_0$ and $\mathcal{D}_1$ of the $\alpha$-filtration for our considered point cloud data $\boldsymbol{X}$ in Figure 9 (plotted on top of each other, which is common practice in topological data analysis). Note that all points in $\mathcal{D}_0$ have 0 as a first coordinate. This corresponds to the fact that all connected components are born at time $\alpha = 0$ in the $\alpha$-filtration. At this time all points are added but no edges between them. $\mathcal{D}_0$ also has exactly one point equal to $(0, \infty)$ plotted on top of the diagram. This quantifies that the simplicial complex eventually constitutes one connected component that never dies, as illustrated by Figure 9. Birth times for points in $\mathcal{D}_1$ are nonzero, since edges are necessary for cycles to be present. They all have finite death-times, since they must be filled in at least at some point in time, as illustrated by Figure 9.

We see that in the combined diagrams, there are six prominently elevated points $(b, d)$ for which the *persistence*—measured by the vertical distance $d - b$ from the point to the diagonal—is significantly higher than all other points (encircled in blue in Figure 10). These quantify the prominent topological features in the model underlying the data $\boldsymbol{X}$ which we discussed earlier. For $\mathcal{D}_0$, these are the four connected components, one for each one of the letters 'I', 'C', 'L', and 'R'. Note that the single point $(0, \infty)$ on top may give a deceptive view that there is only one prominently elevated point in $\mathcal{D}_0$. However, there are as many points in $\mathcal{D}_0$ as data points in $\boldsymbol{X}$ (which may be difficult to deduce from Figure 10 due to overlapping points), and the four encircled points in $\mathcal{D}_0$ are indeed significantly more elevated. For $\mathcal{D}_1$, the two prominently elevated points quantify the hole in the 'R' and the hole between the letters 'C' and 'R'. Note that the birth time of the hole between the letters 'C' and 'L' is much later, as it occurs on a larger scale than the hole in the letter 'R'. The latter occurs at the scale where the four clusters, one for each letter, occur as well (Figures 9 & 10).

**Computational Cost of Persistent Homology**   A remaining disadvantage of persistent homology is its computational cost. Although it is an unparalleled tool for quantifying all from the finest to coarsest topological holes in data, it relies on algebraic computations which can be costly for larger sized simplicial complexes. In the worst case, these computations are cubic in the number of simplices (Otter et al., 2017).

For a data set $\boldsymbol{X} \subseteq \mathbb{R}^d$ of $n$ points, the $\alpha$-complex has size and computation complexity $\mathcal{O}\left(n^{\lceil \frac{d}{2} \rceil}\right)$ (Otter et al., 2017; Toth et al., 2017; Somasundaram et al., 2021). However, this can be significantly lower in practice, for example, if the points are distributed nearly uniformly on a polyhedron (Amenta et al., 2007). Another popular type of filtration is the Vietoris-Rips filtration—for which topological optimization is also implemented by Brüel-Gabrielsson et al. (2020)—which has size and computation complexity $\mathcal{O}(\min(2^n, n^{k+1}))$, where $d \geq k$ is the homology dimension of interest (Otter et al., 2017; Somasundaram et al., 2021). In practice, the choice of $k$ will also significantly

reduce the size of the $\alpha$-complex prior to persistent homology computation. However, the full complex, i.e., the Delaunay triangulation, still needs to be constructed first (Boissonnat et al., 2018). This is often the main bottleneck when working with $\alpha$-complexes of high-dimensional data.

In practice, $\alpha$-complexes are favorable for computing persistent homology from low dimensional point clouds, whereas fewer points in higher dimensions will favor Vietoris-Rips complexes (Somasundaram et al., 2021). Within the context of topological regularization and data embeddings, we aim to achieve a low dimensionality $d$ of the point cloud embedding. This justifies our choice for $\alpha$-filtrations in the main paper.

**Computational Cost of Topological Loss Functions** When one makes use of a sampling fraction $0 < f_{\mathcal{S}} \leq 1$ along with $n_{\mathcal{S}}$ repeats, the computational complexity of the topological loss function reduces to $\mathcal{O}\left(n_{\mathcal{S}}\left((f_{\mathcal{S}}n)^{3\lceil \frac{d}{2}\rceil}\right)\right)$ when using $\alpha$-filtrations, where $d$ is the embedding dimensionality. The added benefit of sampling is that some topological models can be even more naturally modeled, as shown in the main paper.

Nevertheless, as discussed above, the computational complexity may often be significantly lower in practice, due to constraining the dimension of homology for which we optimize, and that common distributions across natural shapes admit reduced size and time complexities of the Delaunay triangulations (Dwyer, 1991; Amenta et al., 2007; Devillers & Duménil, 2019).

Note that many computational improvements as well as approximation algorithms for persistent homology are already available (Otter et al., 2017). However, their integration into topological optimization (and thus regularization) is open to further research.

## B  Appendix: Supplementary Experiments

In this part of the appendix, we include additional experiments that illustrate the usefulness and effectiveness of topological regularization. Section B.1 includes an additional qualitative evaluation on the Harry Potter network, which shows that its graph embedding topologically regularized for a circular prior is consistent with earlier and independent topological data analysis of this network (Vandaele et al., 2020). In Section B.2, we show how including prior expert topological information into the embedding procedure may facilitate automated pseudotime inference in the real cyclic single-cell expression data presented in the main paper. Finally, in Section B.3 we show how topological regularization reacts to different topological loss functions, either designed to model the same prior topological information, or different and potentially wrong prior information.

### B.1  Additional Qualitative Evaluation on the Harry Potter Network

We considered an additional experiment on the Harry Potter graph obtained from `https://github.com/hzjken/character-network`. This graph is composed of characters from the Harry Potter novel (these constitute the nodes in the graph), and edges marking friendly relationships between them (Figure 11). Only the largest connected component is used. This graph has previously been analyzed by Vandaele et al. (2020), who identified a circular model therein that transitions between the 'good' and 'evil' characters from the novel.

To embed the Harry Potter graph, we used a simple graph embedding model where the sigmoid of the inner product between embedded nodes quantifies the (Bernoulli) probability of an edge occurrence (Rendle et al., 2020). Thus, this probability will be high for nodes close to each other in the embedding, and low for more distant nodes. These probabilities are then optimized to match the binary edge indicator vector. Figure 12a shows the result of this embedding, along with the circular model presented by Vandaele et al. (2020). For clarity, character labels are only annotated for a subset of the nodes (the same as by Vandaele et al. (2020)).

We regularized this embedding using a topological loss function $\mathcal{L}_{\text{top}}$ that measures the persistence of the most prominent 1-dimensional hole in the embedding (see also Table 2 in the main paper). The resulting embedding is shown in Figure 12c. Interestingly, the topologically regularized embedding now better captures the circularity of the model identified by Vandaele et al. (2020), and focuses more on distributing the characters along it. Note that although this previously identified model is included in the visualizations, it is not used to derive the embeddings, nor is it derived from them.

For comparison, Figure 12b shows the result of optimizing the ordinary graph embedding (used as initialization) for the same topological loss, but without the graph embedding loss. We observe that this results in a sparse enlarged cycle. Most characters are positioned poorly along the circular model, and concentrate near a small region. Interestingly, even though we only optimized for the topological loss here, it is actually less optimal, i.e., higher than when we applied topological regularization (see also Table 3 in the main paper). This is a result from the sparsity of the circle, which constitutes a larger birth time, and therefore overall a lower persistence of the corresponding hole.

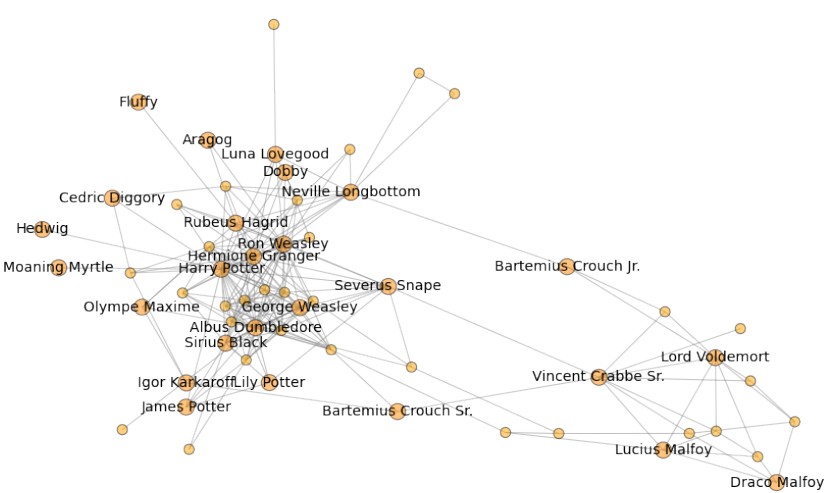

Figure 11: The major connected component in the Harry Potter graph. Edges mark friendly relationships between characters.

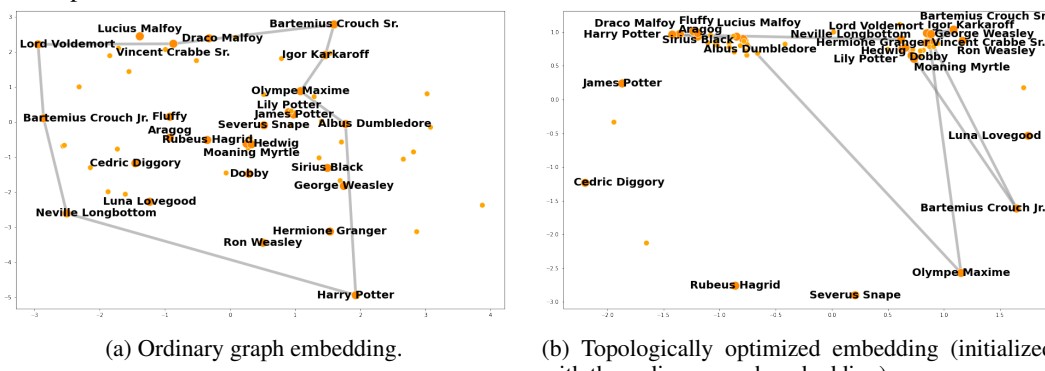

(a) Ordinary graph embedding.      (b) Topologically optimized embedding (initialized with the ordinary graph embedding).

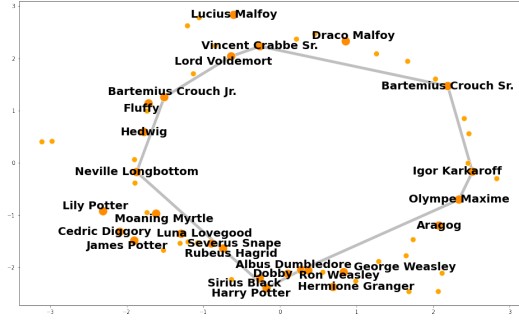

(c) Topologically regularized embedding.

Figure 12: Various embeddings of the Harry Potter graph and the circular model therein.

B.2    APPLICATION: PSEUDOTIME INFERENCE IN CELL TRAJECTORY DATA

Single-cell omics include various types of data collected on a cellular level, such as transcriptomics, proteomics and epigenomics. Studying the topological model underlying the data may lead to better understanding of the dynamic processes of biological cells and the regulatory interactions involved therein. Such dynamic processes can be modeled through trajectory inference (TI) methods, also called *pseudotime analysis*, which order cells along a trajectory based on the similarities in their expression patterns (Saelens et al., 2019).

For example, the *cell cycle* is a well known biological differentiation model that takes place in a cell as it grows and divides. The cell cycle consists of different stages, namely growth (G1), DNA synthesis (S), growth and preparation for mitosis (G2), and mitosis (M). The latter two stages are often grouped together in a G2M stage. Hence, by studying expression data of cells that participate in the cell cycle differentiation model, one may identify the genes involved in and between particular stages of the cell cycle (Liu et al., 2017). Pseudotime analysis allows such study by assigning to each cell a time during the differentiation process in which it occurs, and thus, the relative positioning of all cells within the cell cycle model.

Thus, *the analysis of single cell cycle data constitutes a problem where prior topological information is available*. As the signal-to-noise ratio is commonly low in high-dimensional expression data (Libralon et al., 2009; Zhang et al., 2021), this data is usually preprocessed through a dimensionality reduction method prior to automated pseudotime inference (Cannoodt et al., 2016; Saelens et al., 2019). *Topological regularization provides a tool to enhance the desired topological signal during the embedding procedure, and as such, facilitate automated inference that depends on this signal.*

To illustrate this, we used persistent homology for an automated (cell) cycle and pseudotime inference method, with and without topological regularization during the PCA embedding of the data. The data we used for this experiment is the real cell cycle data presented in the main paper, also previously analyzed by Buettner et al. (2015). The topological regularization procedure we followed here is the same as in the main paper, i.e., we regularized for a circular prior (see also Tables 1 & 2 in the main paper). Our automated pseudotime inference method consists of the following steps.

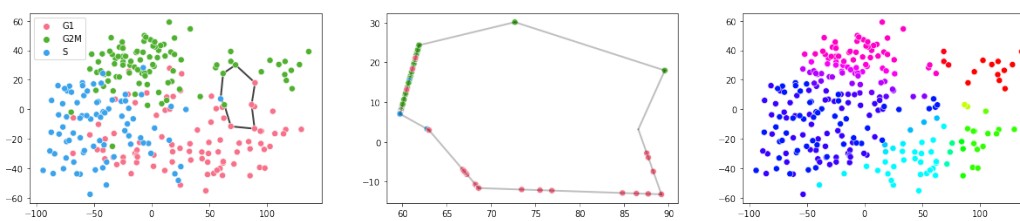

(a) The representation of the most prominent cycle obtained through persistent homology.

(b) Orthogonal projection of the embedded data onto the cycle representation.

(c) The pseudotimes inferred from the orthogonal projection, quantified on a continuous color scale.

Figure 13: Automated pseudotime inference of real cell cycle data through persistent homology, from the PCA embedding of the data.

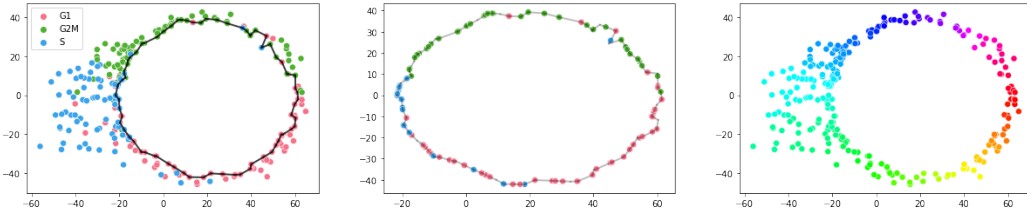

(a) The representation of the most prominent cycle obtained through persistent homology.

(b) Orthogonal projection of the embedded data onto the cycle representation.

(c) The pseudotimes inferred from the orthogonal projection, quantified on a continuous color scale.

Figure 14: Automated pseudotime inference of real cell cycle data through persistent homology, from the topologically regularized PCA embedding of the data.

1. First, a representation of the most prominent cycle in the embedding is obtained through persistent homology from the $\alpha$-filtration, using the Python software library Dionysus (https://pypi.org/project/dionysus/). It can be seen as a circular representation—discretized in edges between data points—of the point in the 1st-dimensional persistence diagram that corresponds to the most persisting cycle (Figures 13a & 14a).

2. An orthogonal projection of the embedded data onto the representative cycle is obtained. This is an intermediate step to derive continuous pseudotimes from a discretized topological representation, as earlier described by Saelens et al. (2019) (Figures 13b & 14b).

3. The lengths between consecutive points on the orthogonal projection are used to obtain circular pseudotimes between $0$ and $2\pi$ (Figures 13c & 14c).

Note that within the scope of the current paper, we do not advocate this to be a new and generally applicable automated pseudotime inference method for single-cell data following the cell cycle model. However, we chose this particular method because it illustrates well what persistent homology identifies in the data, and what we aim to optimize through topological regularization.

For example, we observe that the (representation of) the most prominent cycle in the ordinary PCA embedding is rather spurious, and mostly linearly separates G1 and G2M cells. Projecting cells onto the edges of this cycle places the majority of the cells onto a single edge (Figure 13b). The resulting pseudotimes are mostly continuous only for cells projected onto this edge, whereas they are very discretized for all other cells (Figure 13c). However, by incorporating prior topological knowledge into the PCA embedding, the (representation of) the most prominent cycle in the embedding now better characterizes the transmission between the G1, G2M, and S stages in the cell cycle model (Figure 14a). The automated procedure for pseudotime inference now also reflects a more continuous transmission between the cell stages (Figures 14b & 14c).

## B.3 TOPOLOGICAL REGULARIZATION FOR DIFFERENT LOSS FUNCTIONS

In its current stage, postulating topological prior information is directly performed by designing a topological loss function. This requires some understanding of persistent homology. How to facilitate that design process for lay users is open to further research. A starting point for this is exploring how topological regularization reacts to different (and potentially even wrong) prior topological information and loss functions. This is the topic of this section.

**Different Loss Functions for the same Topological Prior (varying $f_\mathcal{S}$, $n_\mathcal{S}$, or $\tau$)**   In the main paper, we have introduced new topological loss functions (4) and (5). It may be useful to investigate how topological regularization reacts to different choices of the hyperparameters that are used in these loss functions. These are the sampling fraction $f_\mathcal{S}$ and number of repeats $n_\mathcal{S}$ in (4), and the functional threshold $\tau$ in (5). Note that although changing these hyperparameters may result in changing the *geometrical* characteristics that are specified, the *topological* characteristics may be considered to remain roughly the same with respect to the chosen topological loss function.

*Varying $f_\mathcal{S}$ and $n_\mathcal{S}$*   Figure 15 shows the topologically regularized PCA embedding of the real single-cell data from the main paper that follows the cell cycle differentiation model (also discussed in Section 13c) for varying choices of the values $f_\mathcal{S}$ and $n_\mathcal{S}$. The topological loss function and all other hyperparameters are identical to those in the main paper for this data (see also Tables 1 & 2 in the main paper).

We observe that by either increasing $f_\mathcal{S}$ or $n_\mathcal{S}$, the cycle tends to become more defined. More specifically, one can better visually deduce a subset of the data that represents and lies on a nearly perfect circle in the Euclidean plane. Nevertheless, we also conclude that using more data during topological regularization, i.e., higher values of $f_\mathcal{S}$, does not necessarily result in more qualitative embeddings. Indeed, this gives a higher chance of the topological loss function focusing on spurious holes during optimization, for example the hole illustrated in Figure 13a. From Figure 15—although more difficult to deduce due to overlapping points—we see that higher values of $f_\mathcal{S}$ generally result in slightly worse clustering, most notably dispersing the pink colored cells (or, as discussed in Section B.2, G1 cells) more across the entire circular model instead of a confided region.

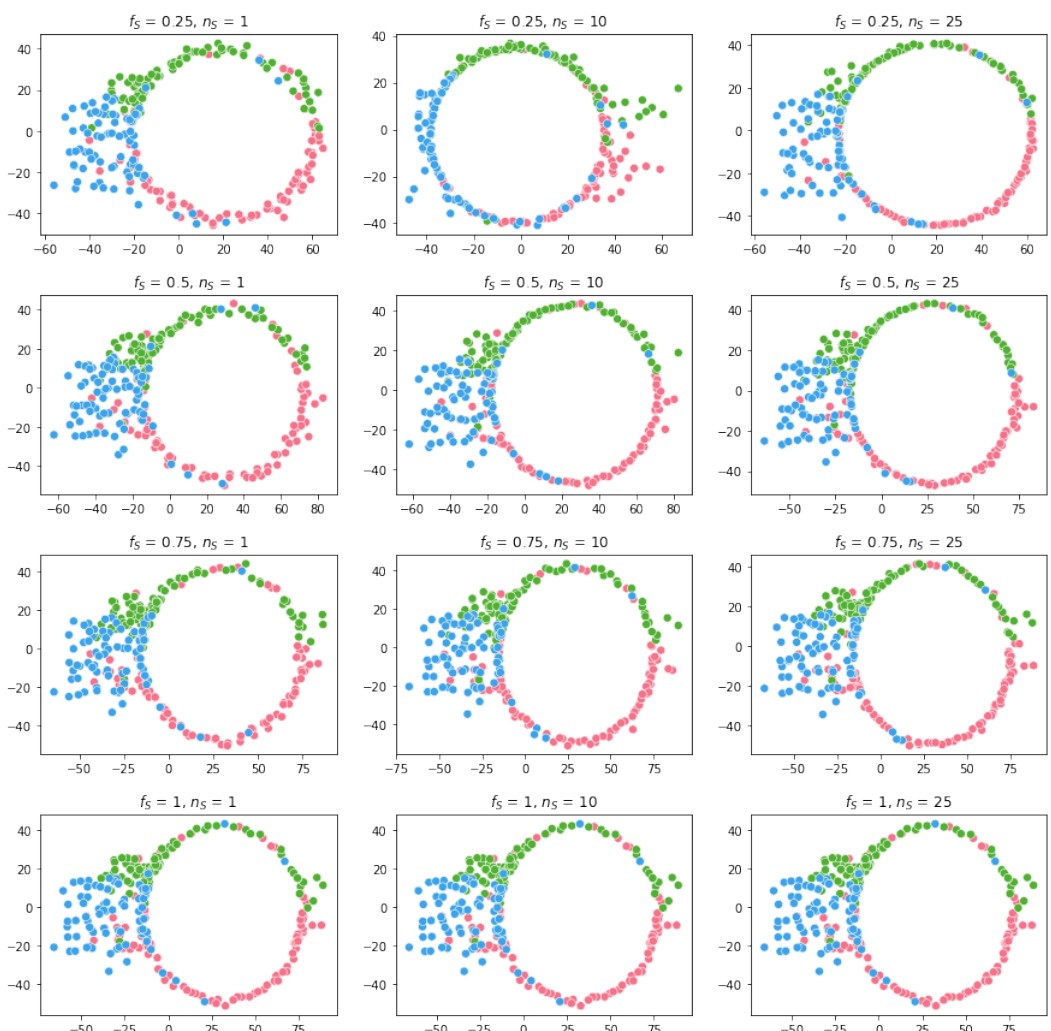

Figure 15: The topologically regularized PCA embedding of the cell cycle data set from the main paper and Section B.2, for varying sampling hyperparameters $f_{\mathcal{S}}$ (the sampling fraction) and $n_{\mathcal{S}}$ (the number of repeats). Note that the embeddings in the bottom row are all identical to $n_{\mathcal{S}} = 1$.

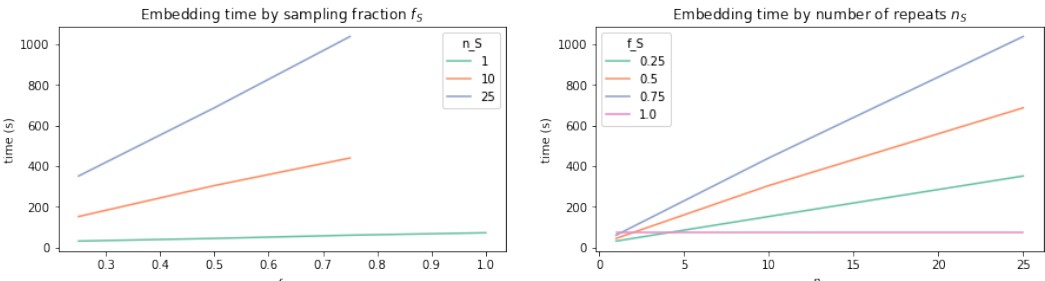

Figure 16: Total computation time of the topologically regularized embedding in seconds (ran for 1000 epochs as in the main paper), by sampling fraction $f_{\mathcal{S}}$ and number of repeats $n_{\mathcal{S}}$. Note that for $f_{\mathcal{S}} = 1$ it is unnecessary to consider $n_{\mathcal{S}} > 1$, and the computation time is constant over $n_{\mathcal{S}}$. Therefore, computation times for $f_{\mathcal{S}} = 1$ and $n_{\mathcal{S}} > 1$ are also not included in the left plot, as they may give a deceptive view. During each step of the topologically regularized embedding optimization, the topological loss function is computed over $264 f_{\mathcal{S}}$ points, repeatedly when specified so by $n_{\mathcal{S}}$.

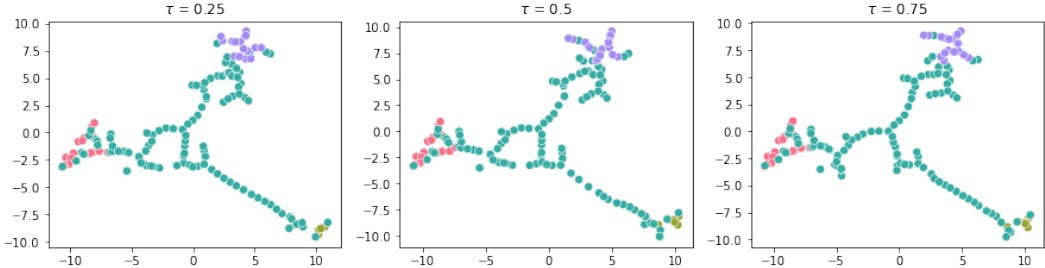

Figure 17: The topologically regularized UMAP embeddings of the bifurcating real single cell data presented in the main paper according to (8), for various functional thresholds $\tau$.

An added advantage of our introduced topological loss function (4), which is approximated through a sampling strategy, is that it increases computational efficiency. One can use lower sampling fractions $f_S$ not only to avoid optimizing for spurious holes, but also conduct this optimization significantly faster, as also discussed in our computational analysis in Section A. This can be seen from Figure 16, which compares the computation time in seconds of the topologically regularized embeddings for different sampling fractions $f_S$ and number of repeats $n_S$.

*Varying $\tau$* Figure 17 shows the topologically regularized UMAP embedding of the real single-cell data following a bifurcating cell differentiation model from the main paper, for varying choices of the functional threshold $\tau$. Recall that $0 \leq \tau \leq 1$ was a threshold on the normalized distance of the embedded points to their mean as defined in (6) in the main paper. Intuitively, $\tau$ specifies how close one looks to the embedding mean, with higher values of $\tau$ allowing more points (thus closer to the embedding mean) to be included when optimizing for three clusters away from the center.

This is consistent with what we observe from Figure 17. While little differences can be observed between $\tau = 0.25$ and $\tau = 0.5$, we observe that for $\tau = 0.75$ (which was the value chosen in the main paper), points near the center of bifurcation are more stretched towards the leaves, i.e., endpoints, in the embedded model. This agrees with the fact that for higher values of $\tau$, more points close to the center are included when optimizing for three separated clusters.

**Different Loss Functions for the same Topological Prior (varying $g$)** For one particular type of prior topological information, one can design different topological loss functions by varying the real-valued function $g$ evaluated on the persistence diagram points that is used in the loss function (2) presented in the main paper. In particular, our topological loss function (3) in the main paper, where we let $g :\equiv b_k - d_k$, is inspired by the topological loss function

$$\mathcal{L}_{\text{top}}(\boldsymbol{E}) := \mathcal{L}_{\text{top}}(\mathcal{D}) = \mu \sum_{k=i, d_k < \infty}^{|\mathcal{D}^{\text{reg}}|} (d_k - b_k)^p \left( \frac{d_k + b_k}{2} \right)^q, \qquad \text{where } d_1 - b_1 \geq d_2 - b_2 \geq \dots,$$

(7)

introduced by Brüel-Gabrielsson et al. (2020), and more formally investigated within an algebraic setting by Adcock et al. (2013). Here, $p$ and $q$ are hyperparameters that control the strength of penalization, whereas $i$ and the choice of persistence diagram (or homology dimension) is used to postulate the topological information of interest. The choice of $\mu \in \{1, -1\}$ determines whether one wants to increase ($\mu = -1$) or decrease ($\mu = 1$) the topological information for which the topological loss function in the data is designed. The term $(d_k - b_k)^p$ can be used to control the prominence, i.e., the persistence, of the most prominent topological holes, whereas the term $\left( \frac{d_k + b_k}{2} \right)^q$ can be used to control whether prominent topological features persist either early or later in the filtration.

Our topological loss function (3) in the main paper is identical to (7) with $p = 1$ and $q = 0$, and the additional change that we sum to the hyperparameter $j$ instead of $|\mathcal{D}^{\text{reg}}|$, which allows for even more flexible prior topological information. Note that this one of the most simple and intuitive, yet flexible, manners to postulate topological loss functions. Indeed, $(d_k - b_k)^{p=1}$ directly measures the persistence of the $k$-th most prominent topological hole for a chosen dimension of interest. The role of $q \neq 0$ is to be further investigated within the context of topological regularization and formulating prior topological information.

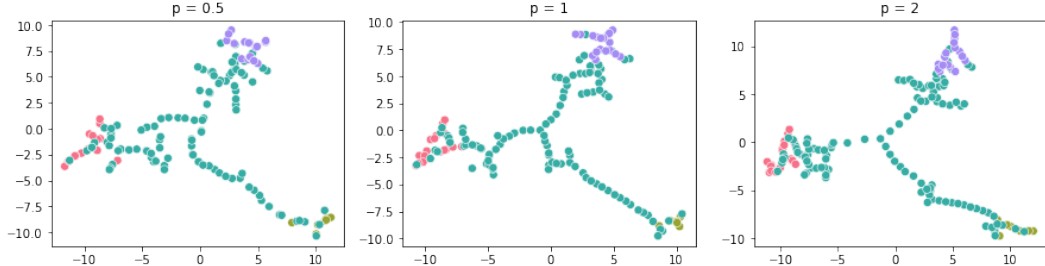

Figure 18: The topologically regularized UMAP embeddings of the bifurcating real single cell data presented in the main paper according to (8), for various values of $p$.

It may already be clear that the same topological information can be postulated through different choices for $p > 0$. To explore this, we considered two other topologically regularized UMAP embeddings of the real bifurcating single cell data presented in the main paper, for the same topological loss function where only the parameter $p$ is varied, i.e.,

$$\mathcal{L}_{\text{top}}(\boldsymbol{E}) = \sum_{k=2}^{\infty}(d_k - b_k)^p - [(d_3 - b_3)^p]_{\mathcal{E}_{\boldsymbol{E}}^{-1}]-\infty,0.75]} . \tag{8}$$

Figure 18 shows the resulting embeddings for $p = \frac{1}{2}$ and $p = 2$, as well as the original embedding for $p = 1$ from the main paper for easy comparison.

In the first term of our topological loss function (8), higher values of $p$ will more strongly penalize larger gaps between neighboring points. This can be seen in Figure 18 *for the majority of the embedded points*, which are more pulled towards the leaves (endpoints) of the represented topological model. However, this results in a few more displaced points near the center of bifurcation, which try to remain connected as we simultaneously fit the UMAP loss function. Indeed, in the main paper we have seen that without the UMAP loss function, the model represented in the topologically optimized embedding will be more fragmented. Hence, in the current experiment, larger values of $p$ in the term $\sum_{k=2}^{\infty}(d_k - b_k)^p$ seem to better motivate many smaller gaps (at the cost of a few larger gaps), whereas lower values appear to motivate more evenly spaced points. The fact that more points are pulled from the center towards the leaves for larger values of $p$ and not from the leaves towards the center, is consistent with the fact that in the original high-dimensional space, more points are far away from each other and the center, than they are close. This has previously been analyzed for this particular data set by Vandaele (2020).

Nevertheless, although different values of $p$ appear to affect the overall spacing between points, we see that the embedded topological shape remains overall recognizable and consistent (Figure 18).

**Different Loss Functions for different Topological Priors (example for PCA)**    As a user may not always have strong prior expert topological knowledge available, it may be useful to study how topological regularization reacts to different topological loss functions that are designed to model weaker or even wrong topological information. Indeed, if the impact of topological regularization on embeddings may be too strong, the user might wrongfully conclude that a particular topological model is present in the data when it can be easily fitted according to *any* specified shape.

To explore this, we considered various additional topological functions to topologically regularize the PCA embedding of the synthetic data set of points lying on a circle with added noise in high dimension, presented in the main paper. Note that here we know that there is exactly one topological model, i.e., a circle, that generates the data. We performed the topologically regularized PCA embedding of this data for the exact same hyperparameters as summarized in Table 1 in the main paper, but now for the three other topological loss functions that are summarized in Table 5. The intuition of these topological loss functions, in order of appearance in Table 5, is as follows.

- The first topological loss function can be considered to specify a weaker form of prior topological information than in the original embedding where we restricted to the term $-(d_1 - b_1)$. Now, the topological loss function states that the sum of persistence of the two

Table 5: Summary of the additional topological losses computed from persistence diagrams $\mathcal{D}$ with points $(b_k, d_k)$ ordered by persistence $d_k - b_k$, used to study the topologically regularized PCA embedding of the Synthetic Cycle from the main paper for weaker or wrong topological information.

| top. loss function | dimension of hole | $f_{\mathcal{S}}$ | $n_{\mathcal{S}}$ |
|---|---|---|---|
| $-(d_1 - b_1) - (d_2 - b_2)$ | 1 | N/A | N/A |
| $-(d_2 - b_2)$ | 1 | N/A | N/A |
| $\sum_{k=2}^{\infty}(d_k - b_k) - [d_3 - b_3]_{\mathcal{E}_E^{-1}]-\infty, 0.75]}$ | 0 - 0 | N/A | N/A |

Figure 19: Various topologically regularized embeddings of the 500-dimensional synthetic data $\boldsymbol{X}$ for which the ground truth model is a circle, as presented in the main paper. (Left to Right) The topological loss function used for topological regularization of the PCA embedding equals the loss function formulated in Table 5 (Top to Bottom).

  most prominent cycles in the embedding should be high. However, this does note impose that the persistence of the second most prominent cycle must necessarily be high.

- The second topological loss function can be considered to specify a partially wrong form of prior topological information. Here, we optimize for a high persistence of the second most prominent cycle in the embedding. Of course, this imposes that the persistence of the most prominent cycle in the embedding must be high as well.
- Our third topological loss function can be considered to specify wrong prior topological information. Here, the topological loss function is exactly the one we used for the bifurcating real single cell data presented in the main paper, or thus, for Figure 18 ($p = 1$) in the Appendix. Hence, this topological loss function is used to optimize for a connected flare with at least three clusters away from the embedding center.

Figure 19 shows the topologically regularized PCA embeddings for these topological loss functions in order. While we observe small changes between the embeddings, which somewhat do agree with their respective designed topological loss function, in all cases, the embeddings do not strongly deviate from the true generating topological model, which is one (and only one) prominent circle.

**Different Loss Functions for different Topological Priors (example for UMAP)**  In case of an orthogonal projection, which we encouraged in Figure 19 as we did in the main paper, it may be difficult to provide an embedding that well captures nontrivial topological models such as in Figure 18, when they are not truthfully present. However, non-linear dimensionality reduction methods such as UMAP may be more prone to model wrong prior topological information, as they directly optimize for the data point coordinates in the embedding space. To explore this, we considered an experiment where we embed the real bifurcating single cell data for a circular prior through the topological loss function $-(d_1 - b_1)$ evaluated on the 1st-dimensional persistence diagram. Thus, we topologically regularized the UMAP embedding of this data set for the exact same topological loss function as we originally used for the synthetic data lying on a circle (see also Figure 4 in the main paper). The result of topological regularization, using the exact same hyperparameters that we originally used for the bifurcating cell trajectory data set as summarized in Table 1 in the main paper, is shown in Figure 20a.

We observe that when the embedding optimization is ran for 100 epochs (as was the embedding in the main paper), a cycle indeed starts to appear in the embedding. To further explore this, we increased the number of epochs to 1000, which did not increase the prominence of the cycle much

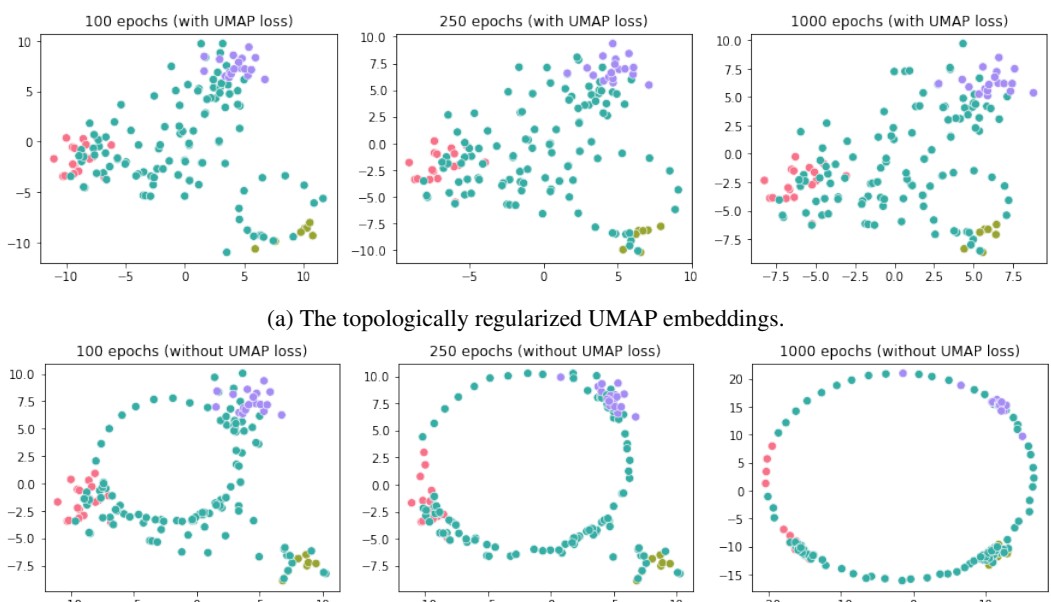

(a) The topologically regularized UMAP embeddings.

(b) Topologically optimized embeddings initialized with the ordinary UMAP embedding from the main paper. Here, the UMAP loss function is not used during topological optimization.

Figure 20: Topologically regularized and optimized UMAP embeddings of the bifurcating cell data, using potentially wrong prior topological (circular) information.

further (Figure 20a). However, by omitting the UMAP loss during the same topological optimization procedure of the embedding, we observe that a cycle becomes much more prominently present in the embedding (Figure 20b). Hence, inclusion of the structural UMAP embedding loss function through topological regularization prevents the cycle to become prominently and falsely represented in the entire data, and the cycle remains spurious in the embedding.

Naturally, higher topological regularization strengths $\lambda_{\text{top}}$ will more significantly impact the topological representation in the data embedding. Thus, caution will still need to be maintained when no prior topological information is available at all. Figure 21 shows the topologically regularized UMAP embeddings for the circular prior for different topological regularization strengths $\lambda_{\text{top}}$. Here, we again ran for 1000 epochs to explore long-term effects of topological regularization during the optimization.

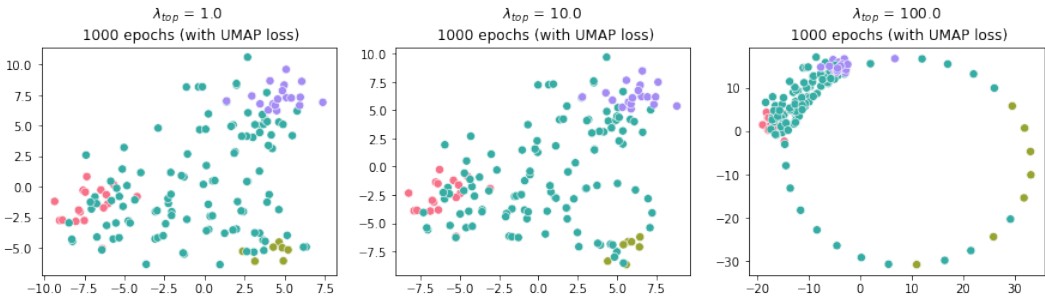

Figure 21: Topologically regularized UMAP embeddings of the bifurcating cell data, using potentially wrong prior topological (circular) information, for various topological regularization strengths $\lambda_{\text{top}}$. For $\lambda_{\text{top}} = 10$, the plot—which is included for easy comparison—is identical to the right plot in Figure 20a.

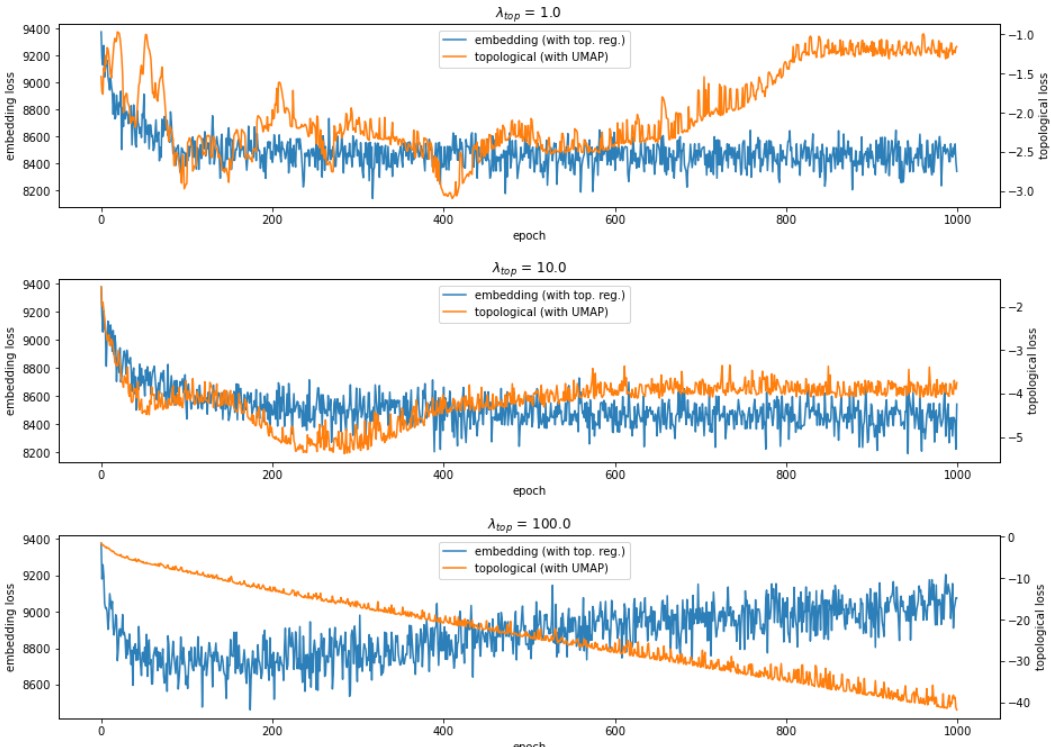

Figure 22: Evolution of the UMAP embedding and topological loss functions during optimization of the topologically regularized embeddings according to the number of epochs, for various topological regularization strengths $\lambda_{\mathrm{top}}$. The regularization strengths are not factored in the plotted topological loss functions for easy comparison. Thus, the topological loss functions directly equal the persistence of the most prominent cycle in each embedding.

When multiplying the original regularization strength $\lambda_{\mathrm{top}} = 10$ with a factor of 10, we see that the circular prior becomes much more prominently present in the topologically regularized UMAP embedding ($\lambda_{\mathrm{top}} = 100$). For lower topological regularization strengths, the topological prior struggles to become prominently represented in the embedded (for $\lambda_{\mathrm{top}} = 10$ as also in Figure 20a), or even appears to have no long-term effect on the embedding at all (for $\lambda_{\mathrm{top}} = 1$). This can also be observed from Figure 22, where the embedding and topological loss functions are plotted according to the number of epochs for the different topological regularization strengths $\lambda_{\mathrm{top}}$. We observe that for $\lambda_{\mathrm{top}} \in \{1, 10\}$, the topological loss functions eventually increase again and stagnate. For $\lambda_{\mathrm{top}} = 100$, the topological loss function appears to decrease indefinitely, at the cost of the embedding loss function which already for smaller numbers of epochs increases again.

Finally, for all three cases we see that the incorrectly specified topological prior is not naturally represented by the data embedding (Figure 21). For example, for $\lambda_{\mathrm{top}} = 100$, the cycle is prominently present, but the majority of points remain clustered together and neglect the circular shape that we aim to model through the topological loss function. This artefact of topological optimization as well as the evolution of the embedding and topological loss functions (which are more generally applicable to higher dimensional embeddings) may provide some visual feedback to the user that the bias to the topological prior may be too weak or too strong. How these observations can be quantified, as to validate the passed topological prior and tune hyperparameters such as the topological regularization strength in an automated manner, is open to further research.

