# OpenReview forum: "Topologically Regularized Data Embeddings"
_ICLR.cc/2022/Conference — ICLR 2022 Poster_

### Official Review · Reviewer_ey6b · 2021-10-28

**Correctness:** 4
**Technical Novelty And Significance:** 3
**Empirical Novelty And Significance:** Not applicable
**Recommendation:** 6
**Confidence:** 3

**Main Review:**

The paper develops a way to construct low-dimensional embeddings that combine an embedding cost (e.g. PCA, t-SNE, UMAP, etc.) with the topological cost, where the latter forces the embedding to have a particular topological structure as represented by the persistent homology. As an example, one can construct a data embedding and force it to be a circle, or a figure-8, or have three clusters.

The paper is based on a series of several 2020-2021 papers, with which I am unfamiliar, so it was difficult for me to judge on the novelty. I found the paper interesting, and the method seems to work well. My biggest criticism is that the goal of the method appears somewhat artificial: low-dimensional embeddings are typically done for data exploration, but one does not want to enforce any particular structure for the purposes of data exploration, and the topological structure is typically not known a priori. That's why I am giving a borderline score.

Most of my comments below are relatively minor requests for clarification.

MAJOR ISSUES

* All considered datasets are very small. Add some discussion or benchmarks of how the runtime scales with dataset size.

* Related work subsection in the Introduction should explain what exactly is the novelty in this paper, compared to Gabrielsson2020, Solomon2021, Carriere2021.


MEDIUM/MINOR ISSUES

* I like Figure 1 and appreciate that the authors attempted to explain persistent homology in the main text. However, I think it could be made clearer. The following things could be briefly explained/mentioned in the text:

a) Panel A -- this looks like kNN graph for various values of k (perhaps alpha=k). I think it's not a kNN graph, but mention something about it.

b) Panel A -- what gets filled in green? All triangles?

c) Panel B -- should we see 4 clusters corresponding to the letters I,C,L,R? Where are they on the persistent homology? Maybe highlight them somehow.

d) Panel B -- what are some prominent points on this plot? One triangle at death=200? A cluster of triangles in the upper-right corner? Would it make sense to annotate them?

e) Panel B -- maybe it should be on the log-scale? Currently many points are clumped in the low-left.

* Related: would it make sense to have some notation for k-dimensional persistence diagram? Like D_{(k)} or something. I was very confused, until I finally realized that there are several different D's. At least clearly mention on page 3 that there is one separate D object for each dimension up to the space dimensionality minus 1.

* Page 3, "these parameters are entire point clouds X" -- should it be X or E? I think E.

* Page 4: it isn't very clear how to set i,j,mu to achieve a desired result. It slowly becomes more clear as one reads further, but it could be great to explain it from the get go. Perhaps even show in Figure 2 what happens for different choices of i? E.g. can one have 4 clusters?

* Particularly unclear is when mu should be 1 and when it should be -1.

* Page 4: If i=j=2, mu=-1 enforces 2 clusters, then why does i=2,j=inf,mu=1 in the example above enforces 1 cluster?

* Page 5, top line: what is n_S? Wasn't defined.

* Section on "flares" -- define "flare".

* Eq (6): very confusing notation in g_E : E -> R : x |-> ||...||

* Figure 3 -- panel (c) looks good but actually the topological loss would be equally happy with any arbitrary splitting of points in two clusters. So the fact that blue points are separated from orange points in panel (c) has nothing to do with the topological loss... this makes this figure and the surrounding text somewhat misleading.

* Page 6: "better captures the circular hole (with L_\bot(W)..." -- but L_\bot has nothing to do with holes, it's the orthogonality term. Should it be L_top? The same confusion appears several times in sections 3.1 and 3.2.

* Section 3.2, 1st dataset -- if it's data on cell differentiation, then why should it be a circle?

* Page 7, Figure 7b: why does Fig 7b look like it does? The topological loss enforces single cluster but here we see multiple clusters. That's confusing.

* Harry Potter dataset should be mentioned in the main text, in Section 3.2.

* Page 9: "we repeated this entire experiment 100 times" -- unclear where the variance comes from.

* Page 9: "we observe that topological regularization consistently..." -- looking at Table 4, the difference between ordinary embedding and topological regularization is much smaller than the standard deviation over repetitions, for three out of four datasets. I think this should be acknowledged in the text.

* Delaunay is consistently spelled wrong.

**Summary Of The Paper:**

The paper develops a way to construct low-dimensional embeddings that combine an embedding cost (e.g. PCA, t-SNE, UMAP, etc.) with the topological cost, where the latter forces the embedding to have a particular topological structure as represented by the persistent homology. As an example, one can construct a data embedding and force it to be a circle, or a figure-8, or have three clusters.

The paper is based on a series of several 2020-2021 papers, with which I am unfamiliar, so it was difficult for me to judge on the novelty. I found the paper interesting, and the method seems to work well. My biggest criticism is that the goal of the method appears somewhat artificial: low-dimensional embeddings are typically done for data exploration, but one does not want to enforce any particular structure for the purposes of data exploration, and the topological structure is typically not known a priori. That's why I am giving a borderline score.

**Summary Of The Review:**

The paper is based on a series of several 2020-2021 papers, with which I am unfamiliar, so it was difficult for me to judge on the novelty. I found the paper interesting, and the method seems to work well. My biggest criticism is that the goal of the method appears somewhat artificial: low-dimensional embeddings are typically done for data exploration, but one does not want to enforce any particular structure for the purposes of data exploration, and the topological structure is typically not known a priori. That's why I am giving a borderline score.

---

> ### Author Response · Authors · 2021-11-22
> **Reaction and actions to reviewer 4 (part 3)**
>
> **Q: Random splitting in Figure 3**
>
> A: This is true. Indeed, since we are not incorporating any other structural information such as we do through embedding losses, this separation is purely a coincidence, and an artefact of the fact that the initialization is ‘okayish’ with respect to the coloring. This distracts from the entire methodology we propose in the paper. We have thus omitted the coloring in Figure 3 in Section 2.2, and thank you for observing this small but important issue.
>
> **Q: circular hole with L_\bot(W)**
>
> A: This is admittedly a poor placement of L_\bot(W) due to space limitations. It has nothing to do with the topological loss. We just wanted to make the reader aware that the projections do not deviate much from orthogonal transformations, which we found rather important. Indeed, through simple linear transformations, we observed that it becomes rather easy to embed ‘small n large p’ data in a perfect circle, but we found this to lead to more unnatural embedding representations.
>
> We have now improved the corresponding paragraphs in the synthetic and cell cycle experiments sections for clarity. We thank you for remarking this, as this might have confused the reader.
>
> **Q: Circular cell differentiation process**
>
> A: This is real data derived from cells in the ‘cell cycle’ differentiation model that takes place in biological cells as they grow and divide. This actually constitutes a problem where prior topological information is available and topological regularization improves automated inference. This has now been illustrated and discussed through our domain application (Appendix B.2).
>
> **Q: Topological cluster encourages single cluster but multiple present**
>
> A: The term that is used to encourage connectedness in the represented model in Figure 7, mathematically optimizes for many small gaps between points that are close. Here, topological optimization indeed achieves this for ‘the majority of the gaps’, at the cost of a few larger gaps that result in the fragmented representation. Topological regularization adds additional pulling and pushing forces that resolve this issue.
>
> We have now analyzed and discussed this more thoroughly through our supplementary experiments in Appendix B.3 (experiments for varying g). Thank you for observing this.
>
> **Q: Mentioning Harry Potter network in text**
>
> A: All supplementary experiments in the Appendix, which now include not only the Harry Potter graph experiment (Appendix B.1) but other experiments as well (Appendix B.2 & B.3), have now been formally described and referred to at the start of the Experiments section of the main paper.
>
> **Q: Unclear where variances come from**
>
> A: These are computed over the random 100 train-test splits from which we obtained performances. This has now been clarified in the discussion of these quantitative results in the Experiments section. Thank you for observing this.
>
> **Q: Little improvement for classification experiments**
>
> A: This is true and has now been acknowledged and clarified in the discussion of these quantitative results in the Experiments section. Thank you for this suggestion.
>
> **Q: Delaunay is spelled wrong**
>
> A: We have now carefully checked and corrected all spellings of ‘Delaunay’ in the paper. Thank you for observing this.

---

> > ### Comment · Reviewer_ey6b · 2021-11-24
> > **Much improved!**
> >
> > I thank the authors for putting in a lot of work to carefully address my criticisms. The paper became much clearer and much easier to read!
> >
> > I still have the same major concern about the goal of the method being somewhat artificial, but I recognize that this criticism is somewhat un-addressable. Given a lot of technical improvements to the presentation, I am willing to withhold this concern and increase my score from 6 to 7: I think the paper may be useful for the field.
> >
> > One minor thing regarding time complexity analysis in the Appendix: it's good to have this theoretical analysis, but what I would like to see is some empirical runtime measurements. Your Cell Cycle dataset had n=264 and it took 35 seconds (Table 2; by the way, it's not mentioned what kind of hardware this time measurement refers to). Would you be able to embed it if it had 2 thousand points? 20 thousand? 200 thousand? I'd like to see some idea of that. Perhaps simulating some data can give an idea of runtimes.

---

> > > ### Author Response · Authors · 2021-11-29
> > > **Thank you! + computational benchmarks**
> > >
> > > Thank you for reconsidering your evaluation in a positive manner, and finding our paper interesting and relevant for publication at ICLR.
> > >
> > > Considering your final comments: all experiments were performed using Python code on a machine equipped with an Intel R© CoreTM i7 processor at 2.6GHz and 8GB of RAM. Thank you for remarking this, we will be sure to add this to the final version of the paper.
> > >
> > > We have now also re-applied topological regularization to our ‘Synthetic Cycle’ data, where we could easily control the number of data points. For 2k points, this took around 4 minutes. For 20k points, this took around 47 minutes. This is without applying sampling. Note that this suggests linear  scaling for this particular embedding method, topological prior, and data; and linear scaling was also empirically observed in Figure 16 in Appendix B.3. While we believe these are encouraging results for practical scalability, we cannot back this up theoretically, so we chose not to emphasize this in the paper as the computational cost of topological regularization may be highly dependent on the type of topological prior, the embedding method, and the nature of the data.

---

> ### Author Response · Authors · 2021-11-22
> **Reaction and actions to reviewer 4 (part 2)**
>
> **Q: alpha-complex vs kNN graph**
>
> A: Unfortunately, unlike many other types of combinatorial structures such as kNN graphs, it is difficult to provide a concise definition and interpretation of an alpha-complex. However, we confirm that similar to the k in kNN graphs, alpha can be considered somewhat to determine the scale at which points become connected. We now emphasized this more clearly, and also directly referred to its formal definition in the figure caption, which is now more emphasized in the background section on persistent homology (in Appendix A, Definition A.0.1). Thank you for this suggestion.
>
> **Q: Triangles get filled in in green**
>
> A: Intuitively, the empty spaces surrounded by green get filled in in green, which we now mentioned more clearly in the figure caption. Thank you for observing this.
>
> **Q: 4 clusters for ‘I’, ‘C’, ‘L’, ‘R’**
>
> A: Indeed, these can be seen as the four more prominently elevated points in the 0-th dimensional persistence diagram. We have now mentioned this in the caption of Figure 1, and also more thoroughly discussed this in Appendix A where the same data is now used as a working example. We also annotated these prominent clusters (as well as the other prominent holes) on the diagrams in Figure 1. Thank you for this suggestion, as this intuition is of major importance for our method.
>
>
> **Q: Annotate prominent points**
>
> A: In reaction to the question above, we anticipate that this will now be more clear.
>
> **Q: Diagrams on log-scale**
>
> A: This visualization is honestly very common in topological data analysis, where the points clumped near the diagonal do not indicate prominent topological features underlying the data. We anticipate that this will now be more clear in reaction to the questions above.
>
> **Q: Different notation for different diagrams**
>
> A: This is very true and actually common practice in topological data analysis. We have now made this more clear that these are two diagrams in the figure caption, which admittedly was essential for our paper in the first place. We thank the reviewer for this helpful suggestion.
>
> **Q: E instead of X**
>
> A: We agree that for consistency this should
>
> **Q: Role of mu**
>
> A: Mu in {1,-1} is a factor used to determine whether we want to minimize or maximize the topological loss function. In our context, this means that for mu = 1, holes (cluster, cycles, …) will be less prominent (as we decrease their persistence), whereas for mu = -1 they will become more prominent. We have now explained this more clearly from the get go in Section 2.1, right above our introduction of the topological loss function (3). Thank you for this suggestion.
>
> Unfortunately, due to space limitations, we find ourselves unable to include the additional suggested experiments in the paper.
>
> Nevertheless, we confirm that we can in fact optimize the ‘ICLR’ data set for four clusters, very similar to how we proceeded for Figure 3. If you are interested in this, you can take a look at the “Optimize ICLR (4 clusters).html” file in the “Output” folder in our supplementary files provided with this submission, and directly explore the results of this.
>
> **Q: mu = 1 vs mu = -1**
>
> A: This has been addressed in reaction to the question above.
>
> **Q: 2 clusters vs 1 cluster**
>
> A: i = j = 2, means that we are looking exactly at the point in the persistence diagram that captures the second most prominent gap in the data, this being the finite gap between the two most prominent clusters in the data. Mu = -1 then specifies that you want this gap to be large.
>
> i = 2, j = infty, means that we are looking at all possible gaps in the data. Note that i = 1 is omitted since the first gap is always infinite by convention. Mu = 1 specifies that we want to decrease the prominence of all these gaps, or thus, make them all smaller, or thus, bring all points closer to each other. This explains why we optimize for 1 cluster.
> Note that we could achieve something similar by i = j = 2 and mu = 1. However, j = infty simply considers many more gaps, thus allowing more movement, per epoch.
>
> **Q: what is n_S**
>
> A: It is the number of times we repeat the sampling to approximate the expected loss. We have now mentioned this more clearly by adding ‘repeats’ before ‘n_S’ in section 2.2. Thank you for observing this.
>
> **Q: define flare**
>
> A: Intuitively, these are ‘star-structured shapes’, which we now mentioned more clearly in Section 2.2 below expression (6). Thank you for observing this.
>
> **Q: Confusing notation in Equation (6)**
>
> A: We have now replaced this by a more intuitive expression in Section 2.2. Thank you for this suggestion.

---

> ### Author Response · Authors · 2021-11-22
> **Reaction and actions to reviewer 4 (part 1)**
>
> Thank you for acknowledging that our paper is interesting and that the method works well.
>
> We anticipated that the fact that topological information is usually unavailable would be one of the main concerns. However, there are indeed applications for which prior topological information is available, and the necessity of a method that can incorporate this into embeddings, as we propose in the current paper, exists. Furthermore, even when no topological prior information is available, we believe that topological regularization may still prove to be useful within an exploratory data analysis context.
>
> We will clarify these and all further comments and suggestions below one-by-one.
>
> **Q: Topological structure is typically not known a priori**
>
> A: We agree that in data exploration tasks there may be no prior topological information available.
>
> However, there are in fact applications where the presence of a topological model is either assumed or known to be present, for example, in biological cell trajectory inference. Here, the data is commonly very noisy. Automated inference methods heavily rely on dimensionality reduction methods to increase the topological signal-to-noise ratio. For such applications, topological regularization provides a way to improve the topological signal during the dimensionality reduction process. We have now included a domain application that shows this, which emphasizes both the necessity and effectiveness of our method (Appendix B.2).
>
> Furthermore, we now include additional experiments that show that when the bias towards the topological prior is wrong or too strong, this may be suggested by some visual feedback to the user (final subsection of Appendix B.3). This suggests future applications even where little to no prior topological information is available.
>
> We thank you for this helpful observation, and hope that you agree that topological regularization can still be useful in many machine learning applications.
>
> **Q: small sized data sets**
>
> A: We observed that the lack of topological representation is a notable problem in ‘small n large p’ data sets. For example, in the case of our synthetic cycle data, PCA would simply recover the circular model on its own when increasing the data size, and topological regularization would not be required. Nevertheless, ‘small n large p’ studies and applications are also important to various communities as well, as for example justified by our real single cell data experiments. We have now emphasized this more clearly in the Experiments section where the synthetic cycle data is introduced. Thank you for observing this.
>
> We now also include a formal computational time analysis of evaluating topological loss functions (end of Appendix A, also formally referred to in Section 2.2 after having introduced our topological loss based on sampling). Additionally we studied and visualized the effect of the sampling fraction, which is linearly related to the data size, on the embedding time (Appendix B.3, Figure 16). Thank you for these helpful suggestions.
>
> **Q: Further explain differences to Gabrielsson2020, Solomon2021, Carriere2021**
>
> A: Simply put, the above work has shown that we can do topological optimization in various settings, formalized its mathematical foundation, and developed the Pytorch/TensorFlow software that allows us to perform topological optimization.
>
> The novelty in our paper lies in
> -	the particular study of topological optimization for point cloud data embedding applications, also addressing its issue of neglecting structural information such as neighborhoods,
> -	introducing new classes of loss functions that work well but also shown to be necessary in our paper,
> -	our many experiments that show the versatility and potential of topological regularization,
> -	and finally, the additional insights that we provided into how noise affects the topological quality of the embedded  model representation, and how topological regularization may accommodate for this.
>
> We have now pointed out the main differences more clearly in a new paragraph in the Related work section. We thank you for this suggestion.
>
> **Q: Making persistent homology more clear**
>
> A: Persistent homology is somewhat of a specialist topic. For this reason, we chose for a mainly visual explanation of persistent homology in Figure 1, as it is otherwise difficult to formally introduce in a concise manner.
>
> Nevertheless, we also agree that there were still improvements that could be made to our current attempt of visually explaining persistent homology, and we have followed your suggestions, as we explain below.
>
> Note that we now also significantly extended the background section on persistence homology in Appendix A.

---

### Official Review · Reviewer_fHUg · 2021-11-01

**Correctness:** 3
**Technical Novelty And Significance:** 2
**Empirical Novelty And Significance:** 2
**Recommendation:** 5
**Confidence:** 3

**Main Review:**

Strengths:

1.	Topological loss is used to regulate low-dimensional embedding method for incorporating prior topological knowledge
2.	Several novel topological loss functions are presented including k-dimensional holes and flares. And the combinations of these basic losses can be applied.


Weaknesses:
1.	It seems that the loss (1) is an incremental formulation by combining existing low-dimensional embedding methods and the simplified topological loss function (2). From (3), the topological loss of this paper is constructed from the embeddings instead of the input data. However, this may be thought as a straightforward extension.
2.	Authors did not explain why only the regular part of (2) is used. It looks like only partial power of the topological optimization is considered by this work. Some explanation for choosing only the regular part is needed.
3.	The claim that “a topological model that is naturally present in the data should be represented well by many subsets of the data” is not quite convincing. Taking the clustering problem as an example, this claim might not true if there are many small clusters when uniform sampling is used. Since it Is the key assumption for (4), authors might want to specify the limitation of the loss (4). Even for the experiments using (4), the analysis on the impact of f_s and n_s is not studied.
4.	Experiments are conducted on both synthetic data and real data, but only the very basic approaches are compared. It is more like the ablation study of the proposed model. Authors need to compare the results obtained by the proposed model with the state-of-the-art methods. At least, the results from the paper using the same real data should be reported as the most relevant baselines.
5.	From the visualization results, it seems that top. optimization embedding is very similar to the top. regularized embedding on most of the data sets expect Figure 7. Does it mean top. loss is sufficient enough to obtain relatively good results?


**Summary Of The Paper:**

Authors of this paper aim to integrating different prior topological knowledge into low-dimensional embeddings, where new set of topological losses are introduced as the topological regularizations to regulate embedding models.

**Summary Of The Review:**

Authors proposed to joint optimize low-dimensional embedding and topological loss, but the proposed method might be thought as a straightforward extension of existing work. The experiments lack the relevant methods for comparisons and the proposed method seems not significantly better than one of simple baselines from visual perspectives. Some claims need to be clarified in detail.

---

> ### Author Response · Authors · 2021-11-22
> **Reaction and actions to reviewer 3 (part 2)**
>
> **Q: Using only the regular part of the diagram**
>
> A: We do not use the essential part simply because we cannot change it through topological optimization of point clouds. The essential part will always be {(0, infty)} for 0-dimensional homology,  since one connected component will eventually persist. For higher dimensional homology, all holes will eventually be filled, and the essential part will always be empty.
>
>  This has now been clarified in the start of Section 2.1. Thank you for suggesting this.
>
> **Q: A topological model that is naturally present in the data should be represented well by many subsets of the data**
>
> A: We agree that this may not always be the case, and rephrased this sentence in Section 2.2.
>
> We now also include additional experiments where we vary all hyperparameters that are important for our newly introduced topological loss functions (start of Appendix B.3). Thank you for this suggestion.
>
> **Q: Ablation experiments only**
>
> A: We confirm that we mainly considered ablation experiments for comparison. However, to the best of our knowledge, there is no existing method that can take any type of data embedding method and any type of topological prior into account, and produce an embedding, such as topological regularization can. We cannot think of a state-of-the-art approach for comparison that can achieve this in an equally versatile manner, or even a different method for every combination of embedding method, topological prior, and data input type that we consider in this paper. We believe this exactly adds to the necessity of the method that we propose.
>
> Nevertheless, if you have any suggestions of state-of-the-art methods which we can include for comparison, we will gladly take them into account for our final version. Thank you for remarking this.
>
> **Q: Only significant differences in Figure 7**
>
> A: We would argue that the differences between topological optimization of the embedding and topological regularization of the embedding are mostly insignificant for the PCA dimensionality reduction. This can be explained by the fact that here, we optimize for the projection matrix W, and not for the point cloud directly such as for example UMAP. As an orthogonal transformation, it will be much more unlikely to capture the presence of non-trivial shapes (say a bifurcating mode) when they are not naturally present in the data. These differences have now been pointed out and discussed through the additional experiments in the final subsection of Appendix B.3. Thank you for observing this.

---

> ### Author Response · Authors · 2021-11-22
> **Reaction and actions to reviewer 3 (part 1)**
>
> Thank you for acknowledging the proposed method as well as the novel topological loss functions.
>
> As we now tried to make more clear in the Related Work section of the paper, our work indeed builds on recent existing work. Nevertheless, we believe that there are many novel contributions that should be considered for publication.
>
> We will address these as well as other specific concerns below one-by-one.
>
> **Q: Straightforward extension of existing work**
>
> A: We agree that we build on recent work in our paper. Nevertheless, we tried to be maximally transparent about this, for example, by formally citing the references on which we base our topological loss functions, but also where topological optimization is analyzed, implemented, and introduced, and from where we obtained and adapted our code.
>
> Needless to say, to the best of our knowledge, topological optimization had not yet been studied nor applied within the context of data embedding. This has now been mentioned in the Related Work section as well.
>
> Admittedly, the fact that we base ourselves on two existing types of loss functions (embedding and topological) gives the impression that topological regularization is easy to realize. However, this is, in our view, exactly what resulted in its versatility, effectiveness, and usability. Indeed, to the best of our knowledge, there is no existing method that can take any type of data embedding method and any type of topological prior into account, and produce an embedding. This is exactly what topological regularization can do. We strongly believe that the possible impact and elegance of the method outweighs the fact that we relied on existing implementations to realize this. We believe that a conceptually simple insight that can tackle a problem that (without this insight) appears very challenging, is perhaps even more valuable than a conceptually complex idea that achieves the same.
>
> Furthermore, we would like to emphasize that, even if the key insight of combining these two loss functions might be considered conceptually simple, realizing it was by no means trivial. The paper thus contains many more contributions in addition to the mere idea of combining these losses, of which we hope the reviewer will see the merit:
> - The limitations of raw topological point cloud optimization had not yet been addressed. In particular, they neglect all structural information such as neighborhoods. Not only did we address this in the current paper, we also provided a solution through topological regularization.
> - We addressed the limitations of existing topological loss functions, and proposed a new family of topological loss functions to overcome these.
> - We included many experiments that show topological regularization is incredibly versatile in terms of the passed embedding method and topological information. To the best of our knowledge, there is no prior work that extensively studies the integration of embeddings and topological optimization as we do in the current paper.
> - Through topological regularization, we provided additional insights into the performance of ordinary data embedding methods for recovering the topological model.
> - We now also include a domain application in biology that shows this method can indeed be used to improve automated inference in real data where topological information is available (Appendix B.2).
> - We now also include additional experiments where we study how topological regularization reacts to different loss functions, either designed for the same, weaker, or wrong topological information. Furthermore, we explore how this may lead to visual feedback to the user that the bias towards the topological prior may be wrong or too strong (Appendix B.3).
> - To make this work possible, we had to extensively collect, adapt, write, and test code. All this code will be made publicly provided with the paper, including many illustrative tutorials.

---

> ### Author Response · Authors · 2021-11-29
> **Kind request for re-evaluation**
>
> Dear reviewer
>
> Thank you again for having taken the time to thoroughly evaluate our submitted manuscript.
>
> In reaction to this, we went at great lengths to thoroughly address and accommodate every single one of your comments, and  submitted a now much improved version of the paper (as already confirmed by one of the reviewers).
>
> We would very much appreciate you going through our responses to your comments, as well as the significantly updated manuscript in response to them. We hope that you are able to reconsider your evaluation and potentially provide additional feedback for a final version of the paper.
>
> Many thanks in advance for your time and efforts.

---

### Official Review · Reviewer_c3ai · 2021-11-01

**Correctness:** 3
**Technical Novelty And Significance:** 2
**Empirical Novelty And Significance:** 2
**Recommendation:** 3
**Confidence:** 4

**Main Review:**

Strengths: The paper tackles an important and challenging problem faced in dimensionality reduction. The proposed method is simple and justified by existing research. The empirical results demonstrate the method improves qualitative results and emphasise that topological optimisation of the embedding space need not enforce any sensible separation of the data, an important consideration.

Weaknesses: The paper is hampered by a lack of notational clarity in essential sections that makes it difficult to determine the full contributions and impact of the proposed method. The following are questions and suggestions to improve the clarity and content of the paper:
1) The methodology rests on equation (2) which demonstrates how topological information is used by the regularisation term, but the notation is difficult to parse.

a) Do the i and j indices refer to indices of the elements of the ordered persistence diagram? It appears that they do, but this could be emphasised. There appear to be two sets of indices: those that are grouped as essential and those termed regular. The ordering is important for the method, but dropping the inequalities and describing containment would help: k \in D^ess, k \in D^reg.

b) Relatedly, the summations in equation (2) could be reconstructed. The first summation appears unnecessary, as it is effectively neutralised by setting g_ess(.) = 0 for the remainder of the paper. Also, what is an example of this summation that has more than one term? The single connected component the embedding tends to is the only term mentioned. This could be expanded.

c) This is an essential point because the choice of j in the summations appears to be a choice point for the method: how deep into the persistence diagram should we go? Ephemeral components that appear for only one step contribute to the loss, but should they? If these components are treated as appearing and disappearing in the same step, then with the chosen g_reg(.) function, they contribute nothing to the loss and could be excluded from the summation.

d) The choice of j and choice of g_reg will be crucial to the results, but the default choices are not compared with alternatives either in section (2) or in the results. How sensitive is the method to changes in these selections?

e) What is |D|? How is it determined? This is the number of points in the persistence diagram, but this should be expanded upon. Specifically, the discussion in the caption of Figure 1 should be moved to the main text and the terminology used there of (H0) and (H1) could be used to improve the clarity of subsequent sections (Section 2.2 par 1 line 3 refers to the 0-dimensional persistence diagram).

2) What is the complexity of the method and how does it scale with the number of data points and type of persistence diagram? In an iterated procedure, is it the case that the digram must be updated with each update of the embedding?

3) As conceded by the authors, the prior topological information is vital to the method, but it is unclear how it is discovered. Thus, it would be useful to see how the method behaves under situations where the prior information is misspecified, this could be included with the synthetic data experiments.

Minor points:
1) Motivation par 1 sentence starting ‘In other examples…’: this sentence seems to contradict itself and should be justified.
2) Additional editing could improve clarity and make space for the expanded methodology section.

**Summary Of The Paper:**

The paper argues for including prior topological information about the structure between data points in some lower dimensional embedding space with the intention of improving the quality of the embeddings produced by tasks such as dimensionality reduction. The paper accomplishes this by adding a novel topological regularisation term to the standard embedding loss. The topological regularisation term incorporates the persistent homology of filtrations of the embedded data points. How this information is manipulated within the term encodes the prior knowledge the investigator wishes to use, such as the embedding should comprise a single cycle. The paper supports its method an array of qualitative and quantitative analyses on synthetic and real datasets.

**Summary Of The Review:**

As the paper stands, I suggest rejection. The method is interesting and I find the empirical results promising, but the methodology sections need large amounts of reworking to improve clarity and to fully justify and explain the choices made by the authors. The paper would benefit greatly from these changes.

---

> ### Author Response · Authors · 2021-11-22
> **Reaction and actions to reviewer 2 (part 2)**
>
> **Q: Computational complexity**
>
> A: A formal computational complexity analysis has now been provided at the end of Appendix A, which is formally referred to in Section 2.2 after we introduce the topological loss function based on sampling. Thank you for this suggestion.
>
> It is indeed the case that the diagram must be updated during each iteration of the optimization, since points that result in the birth or death of topological holes can suddenly change.
>
> Nevertheless, as now also discussed more thoroughly in the paper (at the end of Appendix A), topological optimization is very new. We anticipate that the many improvements to persistent homology computation which are already available will soon be investigated within the context of topological optimization, especially when motivated by this paper.
>
> **Q: Experiments with misspecified topological information**
>
> A: These along with many other additional experiments have now been included in Appendix B.3. In particular, experiments with and analysis of topological regularization with misspecified topological information start on page 23. Thank you for this suggestion.
>
> **Q: ‘In other examples…’**
>
> A: This sentence in the Motivation section has now been fixed. Thank you for observing this.

---

> ### Author Response · Authors · 2021-11-22
> **Reaction and actions to reviewer 2 (part 1)**
>
> Thank you for your many positive comments regarding the interestingness of our paper, its potential impact, and the empirical results.
>
> We have now significantly improved the clarity of our method in reaction to your comments and suggestions, and also further discussed the main differences with the recent related work on which our method is based more thoroughly.
>
> Specific concerns are further addressed below one-by-one.
>
> **Q:  i and j indices refer to indices of the elements of the ordered persistence diagram**
>
> A: Yes indeed. We now immediately dropped the dependence on D_ess from the start in Section 2.1, since (as now also clarified in Section 2.1), it cannot change or thus be optimized for in our case. This improved the notation and readability of our mathematical expressions from the start. Thank you for this helpful suggestion.
>
> **Q: Reconstructing the summations in equation (2)**
>
> A: In reaction to the previous comment, this has now been improved. We now also included additional experiments where there is more variation in the topological loss function, including summation over more than one cycle (Appendix B.3, Table 5, row 1).
>
> **Q: choice of j in the summations appears to be a choice point for the method**
>
> A: We confirm that our method heavily relies on the choice of the indices in the summation. Indeed, these are of crucial importance for specifying the topological prior.
>
> Considering ephemeral components, we have designed all our topological loss functions in the paper to only consider topological holes that remain in the embedding. We only decrease or increase their prominence in the data. For example, when summing over all points in the 0-th dimensional diagram, unless two points are placed exactly on top of each other, all gaps remain present in the embedding (only their sizes change).
>
> Admittedly, the entire design process of topological loss functions can still be somewhat tedious at this point in time, especially for lay users. As mentioned in the Discussion and Conclusion section, we hope that, motivated by the current paper, future research will facilitate this process.
>
> **Q: changing the choice of g and j**
>
> A: As now also explicitly clarified in Section 2.1 and Appendix B.3 (in the experiments for varying g), our choice of the term ‘d_k - b_k’ (thus g) is because we considered this the most basic and intuitive way to express the topological loss, as it directly measures the prominence of the topological hole(s) of interest in the data.
>
> Nevertheless, we now included additional experiments where there is more variation in the design process of the topological loss function, including other choices of g, hyperparameters, and topological priors (all in Appendix B.3).
>
> **Q: determination of |D|**
>
> A: |D| is indeed the number of points in the persistence diagram. However, the cardinality of D does not play an important role in our paper, and we have omitted this notation for clarity. Thank you for observing this.
>
> **Q: moving the caption of Figure 1 to the main text**
>
> We have chosen a mainly visual introduction to persistent homology as we did not want to distract the reader too much with fundamental concepts from algebraic topology.
>
> Nevertheless, we made additional improvements to this visual. This includes explicit references to the two different diagrams (H0 & H1) in the figure caption, as well as annotation and explanation of the prominent holes that can be identified from the persistence diagrams.
>
> Unfortunately, due to space limitations, we find ourselves unable to provide a thorough description of persistent homology and Figure 1 in the main text. Nevertheless, we have significantly extended our background section on persistent homology from 1 to 4.5 pages, which now completely and formally introduces the necessary concepts of persistent homology through the same example in Figure 1 (this is now our working example throughout Appendix A). We also included a reference to the explicit definition of the alpha-complex in the caption of Figure 1, and referred to the specific diagram (H0) as suggested in Section 2.2. Thank you for these suggestions.
>
> While this was the main comment for which we could not completely accommodate in the revision, we hope that you understand that this was mainly due to space limitations. Nevertheless, we hope that you agree that the necessary concepts of persistent homology now became more clear in the ways that we described above.
>
> If you have any other suggestions for omitting parts from the main paper, or moving them to the Appendix, as to make place for an expanded methodology section, we will gladly take them into account for the final version of this paper.

---

> ### Author Response · Authors · 2021-11-29
> **Kind request for re-evaluation**
>
> Dear reviewer
>
> Thank you again for having taken the time to thoroughly evaluate our submitted manuscript.
>
> In reaction to this, we went at great lengths to thoroughly address and accommodate every single one of your comments, and  submitted a now much improved version of the paper (as already confirmed by one of the reviewers).
>
> We would very much appreciate you going through our responses to your comments, as well as the significantly updated manuscript in response to them. We hope that you are able to reconsider your evaluation and potentially provide additional feedback for a final version of the paper.
>
> Many thanks in advance for your time and efforts.

---

> > ### Comment · Reviewer_c3ai · 2021-11-29
> > **Post-rebuttal Comment**
> >
> > Thank you very much for your detailed responses addressing each of my concerns and the great effort in your revisions. I think the paper has substantially improved. It reads much more clearly and the added theoretical details and experimental explorations enhance its contributions. For these reasons, I am willing to increase my review to 6.

---

### Official Review · Reviewer_8muq · 2021-11-01

**Correctness:** 4
**Technical Novelty And Significance:** 3
**Empirical Novelty And Significance:** 3
**Recommendation:** 5
**Confidence:** 2

**Main Review:**

## Positive aspects of the paper:
- The family of loss functions introduced is a novel contribution that can lead to practical applications in areas that can benefit from better representation embeddings, provided that knowledge about the data topology is available and that easier ways to translate this knowledge into the loss functions described became available.

## Concerns and points for improvement:
- Certain topics and mathematical concepts considered in the paper are not familiar to a large audience of the machine learning community. Also, the paper does a poor job in explaining those concepts and I believe many readers in this community will have a hard time understanding it in its present form, due to lack of background. The authors acknowledged this by adding Appendix A, but many points are not sufficiently detailed in there either.
- Considering that acquiring knowledge about the data topology as well as translating this knowledge into an adequate loss function can both be difficult tasks, I would like to see a more thorough evaluation of how misleading the obtained can be if any of these steps are not performed adequately.


**Summary Of The Paper:**

This paper proposes a topological regularization method for incorporating topological prior knowledge for shaping data embeddings. This is achieved by introducing a new family of loss functions based on the characteristics of persistence diagrams. The results of the empirical evaluation suggest that the embeddings produced with the proposed method better capture the topological aspects of input data, by taking into account the prior topological knowledge.

**Summary Of The Review:**

In its present form, I believe the paper should be rejected due to the lack of background information necessary for a large part of ICLR's audience to understand the paper and its contributions.

---

> ### Author Response · Authors · 2021-11-22
> **Reaction and actions to reviewer 1**
>
> Thank you for acknowledging our novel contributions in this paper.
>
> We have now significantly improved the background section in the paper, as to target both the topological data analysis and broader ICLR communities.
>
> These and further specific concerns are further addressed below one-by-one.
>
> **Q: lack of background information for a large part of ICLR’s audience**
>
> A: To improve self-containedness, we extended our background section on persistent homology from 1 to 4.5 pages (Appendix A). We also included additional insights into persistent homology that are important in the main paper (Figure 1 & Section 2.1).
>
> Nevertheless, we would like to add to this that topological data analysis and in particular persistent homology is inherently a specialist topic which is nonetheless important to the ICLR community. This is justified by recent publications such as:
>
> [1] Rieck, B.et al. Neural persistence: A complexity measure for deep neural networks using algebraic topology. In International Conference on Learning Representations (ICLR, 2019).
>
> [2] Jang, U., Jha, S. & Jha, S. On the need for topology-aware generative models for manifold-based defenses. In International Conference on Learning Representations (ICLR, 2020).
>
> [3] Hu, X., Wang, Y., Fuxin, L., Samaras, D. & Chen, C. Topology-aware segmentation using discrete morse theory. In International Conference on Learning Representations (ICLR, 2021)
>
> and even the Topological Representation Learning workshop at ICLR 2021 that successfully pushed forward the fields of computational differential geometry and topology in machine and representation learning.
>
> We anticipate that with the now improved background section, our paper will appeal better to both the specialist community as well as the broader machine learning audience. We thank you for this helpful comment.
>
> **Q: Experiments when the design process of topological loss functions goes wrong**
>
> A: We now included additional experiments where we vary the design process of the topological loss function for the same, weaker, and wrong prior topological information (Appendix B.3). We also show how visual feedback may be obtained when the bias towards the topological prior may be too strong or wrong (Also in Appendix B.3).

---

> > ### Comment · Reviewer_8muq · 2021-11-29
> > **The paper has improved after revision**
> >
> > I believe the paper has significantly improved as my concerns and most of the concerns of the reviewers have been addressed in the revised version.
> > Therefore, I'm willing to increase my score to 6.

---

> ### Author Response · Authors · 2021-11-29
> **Kind request for re-evaluation**
>
> Dear reviewer
>
> Thank you again for having taken the time to thoroughly evaluate our submitted manuscript.
>
> In reaction to this, we went at great lengths to thoroughly address and accommodate every single one of your comments, and  submitted a now much improved version of the paper (as already confirmed by one of the reviewers).
>
> We would very much appreciate you going through our responses to your comments, as well as the significantly updated manuscript in response to them. We hope that you are able to reconsider your evaluation and potentially provide additional feedback for a final version of the paper.
>
> Many thanks in advance for your time and efforts.

---

### Public Comment · ~Yuri_Smirnov1 · 2021-11-10
**Missing citation**

A comment on missing citation. Since you are using the persistence diagrams of complexes, here is the reference for the paper where they were first introduced, under the name of canonical forms : Barannikov, S.(1994) "The Framed Morse Complex and its Invariants", Advances in Soviet Mathematics, 21: 93–115. Also, the computation of persistence diagrams is based on the algorithm described in section 2.1 of this paper.

---

> ### Author Response · Authors · 2021-11-22
> **Action to missing citation**
>
> **Q: missing citation**
>
> A: The citation has now been added twice to the main paper, at the start of Section 2.1. Thanks for this helpful suggestion.

---

### Decision · Program_Chairs · 2022-01-20

**Decision:**

Accept (Poster)

**Comment:**

This paper proposes loss functions to encode topological priors during data embedding, based on persistence diagram constructions from computational topology.  The paper initially had some expositional issues and technical questions, but the authors did an exceptional job of addressing them during the rebuttal period----nearly all reviewers raised their scores (or intended to but didn't update the numbers on their original reviews).

The AC is willing to overlook some of the remaining questions. For example, concerns that topology isn't well known in the ICLR community (8muq) are partially addressed by the improved exposition (and it's OK to have technically sophisticated papers so long as some reviewers were able to evaluate them).  And, future work can address scalability of the algorithm, which indeed does seem to be a challenge here (ey6b).

In the final "camera ready," the authors are encouraged to address any remaining comments and to consider adding experiments/discussion regarding scalability to larger datasets.